# Online Influence Maximization under Linear Threshold Model

**Shuai Li**[1][*] **Fang Kong**[1] **Kejie Tang**[1] **Qizhi Li**[1] **Wei Chen**[2]
[1]Shanghai Jiao Tong University  [2]Microsoft Research
{shuaili8,fangkong,tangkj00,qizhili}@sjtu.edu.cn  weic@microsoft.com

## Abstract

Online influence maximization (OIM) is a popular problem in social networks to learn influence propagation model parameters and maximize the influence spread at the same time. Most previous studies focus on the independent cascade (IC) model under the edge-level feedback. In this paper, we address OIM in the linear threshold (LT) model. Because node activations in the LT model are due to the aggregated effect of all active neighbors, it is more natural to model OIM with the node-level feedback. And this brings new challenge in online learning since we only observe aggregated effect from groups of nodes and the groups are also random. Based on the linear structure in node activations, we incorporate ideas from linear bandits and design an algorithm `LT-LinUCB` that is consistent with the observed feedback. By proving group observation modulated (GOM) bounded smoothness property, a novel result of the influence difference in terms of the random observations, we provide a regret of order $\tilde{O}(\text{poly}(m)\sqrt{T})$, where $m$ is the number of edges and $T$ is the number of rounds. This is the first theoretical result in such order for OIM under the LT model. In the end, we also provide an algorithm `OIM-ETC` with regret bound $O(\text{poly}(m)\,T^{2/3})$, which is model-independent, simple and has less requirement on online feedback and offline computation.

## 1 Introduction

Social networks play an important role in spreading information in people's life. In viral marketing, companies wish to broadcast their products by making use of the network structure and characteristics of influence propagation. Specifically, they want to provide free products to the selected users (seed nodes), let them advertise through the network and maximize the purchase. There is a budget of the free products and the goal of the companies is to select the optimal seed set to maximize the influence spread. This problem is called influence maximization (IM) [19] and has a wide range of applications including recommendation systems, link prediction and information diffusion.

In the IM problem, the social network is usually modeled as a directed graph with nodes representing users and directed edges representing influence relationship between users. IM studies how to select a seed set under a given influence propagation model to maximize the influence spread when the weights are known. Independent cascade (IC) model and linear threshold (LT) model [19] are two most widely used models to characterize the influence propagation in a social network, and both models use weights on edges as model parameters.

In many real applications, however, the weights are usually unknown in advance. For example, in viral marketing, it is unrealistic to assume that the companies know the influence abilities beforehand. A possible solution is to learn those parameters from the diffusion data collected in the past [6, 36]. But this method lacks the ability of adaptive learning based on the need of influence maximization.

---

[*]Corresponding author

This motivates the studies on the online influence maximization (OIM) problem [28, 10, 11, 47, 49, 50, 45, 44], where the learner tries to estimate model parameters and maximize influence in an iterative manner.

The studies on OIM are based on the multi-armed bandit (MAB) problem, which is a classical online learning framework and has been well studied in the literature [27]. MAB problem is formulated as a $T$-round game between a learner and the environment. In each round, the learner needs to decide which action to play and the environment will then reveal a reward according to the chosen action. The objective of the learner is to accumulate as many rewards as possible. An MAB algorithm needs to deal with the tradeoff between exploration and exploitation: whether the learner should try actions that has not been explored well yet (exploration) or focus on the action with the best performance so far (exploitation). Two algorithms, the explore-then-commit (ETC) [15] and the upper confidence bound (UCB) [4], are widely followed in the stochastic MAB setting, where the reward of each action follows an unknown but fixed distribution.

Most existing works in OIM focus on IC model under edge-level feedback [10, 11, 47, 49, 50], where the information propagates independently between pairs of users and the learner can observe the liveness of individual edges as long as its source node is influenced. The independence assumption makes the formulation simple but a bit unrealistic. Often in the real scenarios, the influence propagations are correlated with each other. The LT model is usually used to model the herd behavior that a person is more likely to be influenced if many of her friends are influenced [7, 17, 20]. Thus for the LT model, it is more natural to use the node-level feedback where we only observe the node activations, since it is hard to pinpoint which neighbor or neighbors actually contribute to an activation in a herd behavior.

In this paper, we first formulate the OIM problem under the LT model with the node-level feedback and distill effective information based on the feedback. The main challenge is that only the aggregated group effect on node activations can be observed and the aggregated groups are also random. Based on the linear structure of the LT model, we incorporate the idea of linear bandits and propose the `LT-LinUCB` algorithm, whose update mechanism is consistent with the distilled information. By proving group observation modulated (GOM) bounded smoothness, a key property on the influence spread under two different weight vectors, we can bound the regret. Such a property is similar to the triggering probability modulated (TPM) bounded smoothness condition under the IC model with edge-level feedback [47], but the derivation in our case under the node-level feedback is more difficult. The regret is of order $O(\mathrm{poly}(m)\sqrt{T}\log T)$, where $m$ is the number of edges and $T$ is the number of rounds. Our `LT-LinUCB` is the first OIM algorithm under the LT model that achieves the regret in this order. Finally we give `OIM-ETC` algorithm, applying to both IC and LT with node-level feedback. Though simple, it has less requirement on the observed feedback and the offline computation, and it achieves the regret bound $O(\mathrm{poly}(m)T^{2/3}), O(\mathrm{poly}(m)\log(T)/\Delta^2)$.

**Related Work**   The problem of IM was first proposed as a discrete optimization problem by Kempe *et al.* [19]. Since then, various aspects of IM have been extensively studied (see [9, 31] for surveys in this area). Two most popular models in this field are the IC and LT models. The former assumes that the influence between pairs of users are independent and the latter characterizes the herd behavior. Some works [46, 18, 19, 42] study the IC model and some [12, 16, 19, 42] study the LT model. They all assume the weights on the edges are known and focus on the model properties and approximated solutions. We treat them as the *offline* setting.

When the weight vectors are unknown, Chen *et al.* [11, 47] study the problem in the *online* setting, selecting seed sets as well as learning the parameters. They study the IC model with edge-level feedback, propose CUCB algorithm and show that CUCB achieves the distribution-dependent and distribution-independent regret bounds of $O(\mathrm{poly}(m)\log(T))$ and $O(\mathrm{poly}(m)\sqrt{T})$ respectively. Later Wen *et al.* [49] consider the large-scale setting and assume the edge probability is a linear function of the edge's feature vector. They provide a LinUCB-based algorithm with $O(dmn\sqrt{T}\ln(T))$ worst-case regret, where $d$ is the feature dimension and $n$ is the number of nodes. Wu *et al.* [50] assume that each edge probability can be decomposed as the product of the influence probability of the start node and the susceptibility probability of the end node motivated by network assortativity. All these works study the IC model with edge-level feedback.

Vaswani *et al.* [44] uses a heuristic objective function for OIM and brings up a model-independent algorithm under the pairwise feedback, where a node is influenced by a seed node or not. This applies to both IC and LT and the feedback scheme is relaxed than the edge-level feedback. Unfortunately,

however, the heuristic objective has no theoretical approximation guarantee. Also, Vaswani *et al.* [45] study the IC model with node-level feedback about the estimation gap to that under the edge-level feedback but has no regret analysis. A report [43] studies the LT model with node-level feedback by optimization approaches but without theoretical guarantees. There is another work [26] studying the problem of linear multi-resource allocation, which can be formulated as a bipartite LT model. But they assume every node in the left partition (resources) is selected and the algorithm needs to assign allocations for each pair of left node and right node (tasks) representing the corresponding allocation of resources on tasks. Thus the problem is different from our OIM. The OIM problem under LT has been open for several years. We are the first to provide a reasonable formulation with an algorithm of regret $\tilde{O}(\sqrt{T})$.

OIM is a variant of combinatorial MAB (CMAB) [10, 22], where in each round the learner selects a combination of (base) arms. Most works [25, 24] study stochastic setting with the linear objective and semi-bandit feedback where the learner can observe the selected base arm's reward and the reward of the action is a linear function of these base arms' rewards. CMAB in the stochastic setting with the linear objective and bandit feedback, where only the linear reward of the selected combination can be observed, is a special case of linear bandits. In the linear bandit setting, the learner selects a vector each round and the reward is a linear function of the selected vector action [3]. The most popular method to solve it is to construct confidence ellipsoids [14, 1, 38]. There are also works [8, 13] for CMAB in the adversarial setting and bandit feedback. But OIM is different: its objective function is non-linear and is dependent on unchosen and probabilistically triggered base arms.

OIM is related to the problem of online learning with graph feedback [2] where the learner can observe the feedback of unchosen arms based on the graph structure. Though some of them study random graphs [33, 29, 21] where the set of observed arms is random, the settings are different. Under the graph feedback, the observations of unchosen arms are additional and the reward only depends on the chosen arms, while under the OIM, the additional observations also contribute to the reward. Cascading bandits [23, 30] also consider triggering on any selected list of arms and the triggering is in the order of the lists. Compared with graph feedback and OIM setting, its triggering graph is determined by the learning agent, not the adversary.

As a generalization of graph feedback, partial monitoring [5] is also related to OIM. Most works in this direction, if applied directly to the OIM setting, are inefficient due to the exponentially large action space. Lin *et al.* [32] study a combinatorial version of partial monitoring and their algorithm provides a regret of order $O(\text{poly}(m)T^{2/3}\log T)$ for OIM with LT. Our OIM-ETC, however, has regret bounds of $O(\text{poly}(m)T^{2/3})$ (better in the order of $T$) as well as a problem-dependent bound $O(\text{poly}(m)\log T)$.

## 2   Setting

This section describes the setting of online influence maximization (OIM) under linear threshold (LT) model. The IM problem characterizes how to choose the seed nodes to maximize the influence spread on a social network. The network is usually represented by a directed graph $G = (V, E)$ where $V$ is the set of users and $E$ is the set of relationships between users. Each edge $e$ is associated with a weight $w(e) \in [0, 1]$. For example, an edge $e = (u, v) =: e_{u,v}$ could represent user $v$ follows user $u$ on Twitter and $w(e)$ represents the 'influence ability' of user $u$ on user $v$. Denote $w = (w(e))_{e \in E}$ to be the weight vector and $n = |V|, m = |E|, D$ to be node number, edge number and the diameter respectively, where the diameter of the graph is defined as the maximum (directed) distance between the pair of nodes in any connected component. Let $N(v) = N^{\text{in}}(v)$ be the set of all incoming neighbors of $v$, shortened as in-neighbors.

Recall that under IC model, each edge is alive with probability equal to the associated weight independently and a node is influenced if there is a directed path connecting from a seed node in the realized graph. Compared to the IC model, the LT model does not require the strong assumption of independence and describes the joint influence of the active in-neighbors on a user, reflecting the herd behavior that often occurs in real life [7, 17, 20].

Now we describe in detail the diffusion process under the LT model. Suppose the seed set is $S$. In the beginning, each node is assigned with a threshold $\theta_v$, which is independently uniformly drawn from $[0, 1]$ and characterizes the susceptibility level of node $v$. Denote $\theta = (\theta_v)_{v \in V}$ to be the threshold vector. Let $S_\tau$ be the set of activated nodes by the end of time $\tau$. At time $\tau = 0$, only nodes in

the seed set are activated: $S_0 = S$. At time $\tau + 1$ with $\tau \geq 0$, for any node $v \notin S_\tau$ that has not been activated yet, it will be activated if the aggregated influence of its active in-neighbors exceeds its threshold: $\sum_{u \in N(v) \cap S_\tau} w(e_{u,v}) \geq \theta_v$. Such diffusion process will last at most $D$ time steps. The size of the influenced nodes is denoted as $r(S, w, \theta) = |S_D|$. Let $r(S, w) = \mathbb{E}\left[r(S, w, \theta)\right]$ be the *influence spread* of seed set $S$ where the expectation is taken over all random variables $\theta_v$'s. The IM problem is to find the seed set $S$ with the size at most $K$ under weight vector $w$ to maximize the influence spread, $\max_{S \in \mathcal{A}} r(S, w)$, where $\mathcal{A} = \{S \subset V : |S| \leq K\}$ is the *action set* for the seed nodes. We also adopt the usual assumption that $\sum_{u \in N(v)} w(e_{u,v}) \leq 1$ for any $v \in V$. This assumption makes LT have an equivalent live-edge graph formulation like IC model [19, 9]. The term of graph $G$ and seed size $K$ will be omitted when the context is clear. Here we emphasize that the model parameters are the weights $w$ while the threshold vector $\theta$ is not model parameter (which follows uniform distribution).

The (offline) IM is NP-hard under the LT model but it can be approximately solved [19, 42]. For a fixed weight vector $w$, let $S_w^{\text{Opt}}$ be an optimal seed set and $\text{Opt}_w$ be its corresponding influence spread: $S_w^{\text{Opt}} \in \text{argmax}_{S \in \mathcal{A}} r(S, w)$ and $\text{Opt}_w = r(S_w^{\text{Opt}}, w)$. Let `Oracle` be an (offline) oracle that outputs a solution given the weight vector as input. Then for $\alpha, \beta \in [0, 1]$, the `Oracle` is an $(\alpha, \beta)$-approximation if $\mathbb{P}\left(r(S', w) \geq \alpha \cdot \text{Opt}_w\right) \geq \beta$ where $S' = \text{Oracle}(w)$ is a solution returned by the `Oracle` for the weight vector $w$. Note when $\alpha = \beta = 1$ the oracle is *exact*.

The online version is to maximize the influence spread when the weight vector (or the model parameter) $w = (w(e))_{e \in E}$ is unknown. In each round $t$, the learner selects a seed set $S_t$, receives the observations and then updates itself accordingly. For the type of observations, previous works on IC mostly assume the edge-level feedback: the learner can observe the outgoing edges of each active node [11, 49, 50]. But for the LT model, it is not very realistic to assume the learner can observe which in-neighbor influences the target user since the LT model characterizes the aggregate influence of a crowd. So we consider a more realistic *node-level feedback*[2] in this paper: the learner can only observe the influence diffusion process on node sets as $S_{t,0}, \ldots, S_{t,\tau}, \ldots$ in round $t$.

The objective of the OIM is to minimize the cumulative $\eta$-scaled regret [10, 49] over total $T$ rounds:

$$R(T) = \mathbb{E}\left[\sum_{t=1}^{T} R_t\right] = \mathbb{E}\left[\eta \cdot T \cdot \text{Opt}_w - \sum_{t=1}^{T} r(S_t, w)\right], \tag{1}$$

where the expectation is over the randomness on the threshold vector and the output of the adopted offline oracle in each round .

Throughout this paper, we will use 'round' $t$ to denote a step in online learning and use 'time' $\tau$ of round $t$ to denote an influence diffusion step of seed set $S_t$ in round $t$.

## 3   `LT-LinUCB` **Algorithm**

In this section, we show how to distill effective information based on the feedback and propose a LinUCB-type algorithm, `LT-LinUCB`, for OIM under LT. For each node $v \in V$, denote $w_v = (w(e_{u,v}))_{u \in N(v)}$ to be the weight vector of its incoming edges. Let $\chi(e_{u,v}) \in \{0, 1\}^{|N(v)|}$ be the one-hot representation of the edge $e_{u,v}$ over all of $v$'s incoming edges $\{e_{u,v} : u \in N(v)\}$, that is its $e'$-entry is 1 if and only if $e' = e_{u,v}$. Then $w(e_{u,v}) = \chi(e_{u,v})^\top w_v$. For a subset of edges $E' \subseteq \{e_{u,v} : u \in N(v)\}$, we define $\chi(E') := \sum_{e \in E'} \chi(e) \in \{0, 1\}^{|N(v)|}$ to be the vector whose $e$-entry is 1 if and only if $e \in E'$. Here we abuse the notation that $\chi(\{e\}) = \chi(e)$. By this notation, the weight sum of the edges in $E'$ is simply written as $\chi(E')^\top w_v$. A subset $V' \subset N(v)$ of $v$'s in-neighbors can activate $v$ if the weight sum of associated edges exceeds the threshold, that is $\chi(E')^\top w_v \geq \theta_v$ with $E' = \{e_{u,v} : u \in V'\}$.

Fix a diffusion process $S_0, S_1, \ldots, S_\tau, \ldots$, where the seed set is $S_0$. For each node $v$, define

$$\tau_1(v) := \min_\tau \{\tau = 0, \ldots, D : N(v) \cap S_\tau \neq \emptyset\} \tag{2}$$

as the earliest time step when node $v$ has active in-neighbors. Particularly we set $\tau_1(v) = D + 1$ if node $v$ has no active in-neighbor until the diffusion ends. For any $\tau \geq \tau_1(v)$, further define

$$E_\tau(v) := \{e_{u,v} : u \in N(v) \cap S_\tau\} \tag{3}$$

as the set of incoming edges associated with active in-neighbors of $v$ at time step $\tau$.

Recall that the learner can only observe the aggregated influence ability of a node's active in-neighbors. Let $\tau_2(v)$ represent the time step that node $v$ is influenced ($\tau_2(v) > \tau_1(v)$), which is equivalent to mean that $v$'s active in-neighbors of time $\tau_2(v) - 1$ succeed to influence it but those in time $\tau_2(v) - 2$ fail ($E_{-1} = \emptyset$). Thus the defintion of $\tau_2(v)$ can be written as

$$\tau_2(v) := \left\{\tau = 0, \ldots, D : \chi(E_{\tau-2}(v))^\top w_v < \theta_v \leq \chi(E_{\tau-1}(v))^\top w_v\right\} . \tag{4}$$

For consistency, we set $\tau_2(v) = D + 1$ if node $v$ is finally not influenced after the information diffusion ends. Then based on the definition of $\tau_1(v)$ and $\tau_2(v)$, we can obtain that node $v$ is not influenced at time $\tau \in (\tau_1(v), \tau_2(v))$, which means that the set of active in-neighbors of $v$ at time step $\tau - 1$ fails to activate it.

According to the rule of information diffusion under the LT model, an event that $E' \subseteq \{e_{u,v} : u \in N(v)\}$ fails to activate $v$ is equivalent to $\chi(E')^\top w_v < \theta_v$, which happens with probability $1 - \chi(E')^\top w_v$ since $\theta_v$ is uniformly drawn from $[0, 1]$. Similarly an event that $E' \subseteq \{e_{u,v} : u \in N(v)\}$ succeeds to activate $v$ is equivalent to $\chi(E')^\top w_v \geq \theta_v$, which happens with probability $\chi(E')^\top w_v$. So for node $v$ who has active in-neighbors, $v$ is not influenced at time step $\tau$ ($\tau_1(v) < \tau < \tau_2(v)$) means that the set of $v$'s active in-neighbors by $\tau - 1$ fails to activate it, thus we can use $(\chi(E_{\tau-1}(v)), 0)$ to update our belief on the unknown weight vector $w_v$; $v$ is influenced at time step $\tau_2(v)$ means that the set of $v$'s active in-neighbors by $\tau_2(v) - 1$ succeeds to activate it, we can thus use $(\chi(E_{\tau_2(v)-1}(v)), 1)$ to update our belief on the unknown weight vector $w_v$; $v$ is finally not influenced means that all of its active in-neighbors (by time step $D$) fail to activate it, we can use $(\chi(E_{\tau_2(v)-1}(v)), 0)$ to update $w_v$ since $\tau_2(v)$ is defined as $D + 1$ in this case. Note all of these events are correlated (based on a same $\theta_v$), thus we can only choose at most one of them to update $w_v$ for node $v$ who has active in-neighbors. If $v$ has no active in-neighbors, we have no observation on $w_v$ and could update nothing.

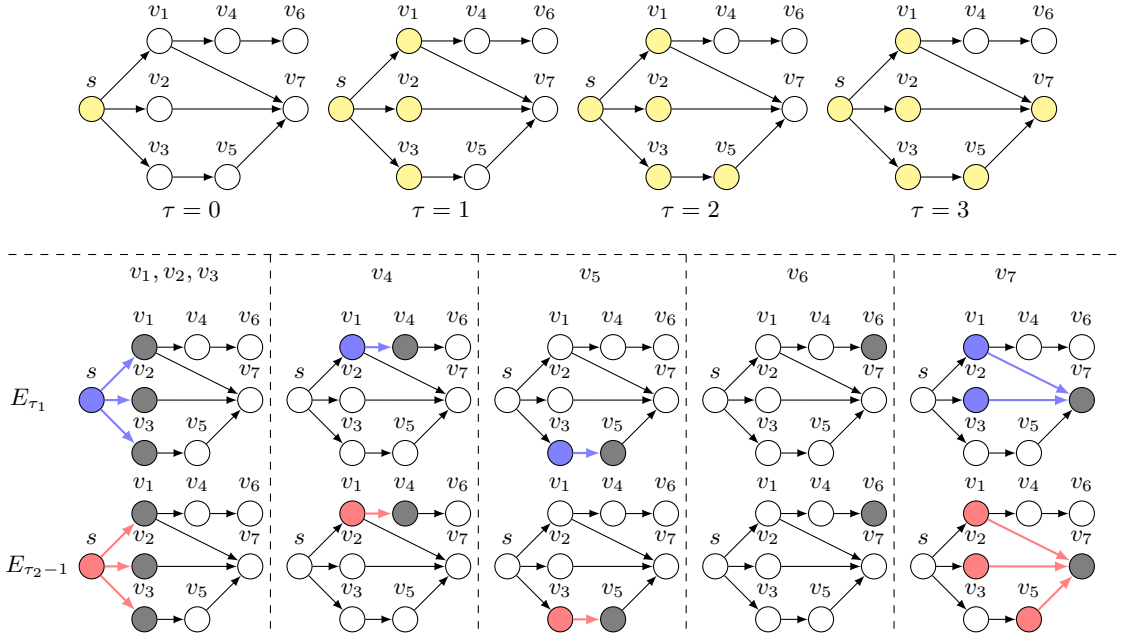

Figure 1: An example of diffusion process starting from $S = \{s\}$ under LT. The upper part describes an influence diffusion process where yellow nodes represent influenced nodes by the current time. The lower part describes what $E_{\tau_1}, E_{\tau_2-1}$ are where we use blue (red) color to represent the edges and the associated active in-neighbors in $E_{\tau_1}$ ($E_{\tau_2-1}$, respectively) for the objective black node.

Figure 1 gives an example of diffusion process and the definitions of edge-sets $E_{\tau_1}$ and $E_{\tau_2-1}$. The diffusion process is illustrated by the upper four figures, where the set $S_\tau$ of influenced nodes by time $\tau$ is yellow colored. The lower five columns represent the sets $E_{\tau_1}, E_{\tau_2-1}$ for different nodes. For example, node $v_7$ has active in-neighbors starting from $\tau = 1$, thus $\tau_1(v_7) = 1$ and $E_{\tau_1(v_7)}(v_7) = \{e_{u,v_7} : u \in N(v_7) \cap S_1\} = \{e_{v_1,v_7}, e_{v_2,v_7}\}$. And $v_7$ is influenced at $\tau = 3$ thus $\tau_2(v_7) = 3$ and $E_{\tau_2(v_7)-1}(v_7) = \{e_{u,v_7} : u \in N(v_7) \cap S_2\} = \{e_{v_1,v_7}, e_{v_2,v_7}, e_{v_5,v_7}\}$. Node $v_6$ has no active in-neighbors, thus $\tau_1(v_6) = \tau_2(v_6) = D + 1$, both its $E_{\tau_1(v_6)}(v_6)$ and $E_{\tau_2(v_6)-1}(v_6)$ are empty sets.

The above describes how to distill key observations for a diffusion under the LT model and also explains the update rule in the design of the algorithm. Denote $\tau_1, \tau_2, E_\tau$ at round $t$ as $\tau_{t,1}, \tau_{t,2}, E_{t,\tau}$ and the diffusion process at round $t$ as $S_{t,0}, \ldots, S_{t,\tau}, \ldots$. Here we abuse a bit the notation $S$ to represent both the seed set and the spread set in a round when the context is clear.

Our algorithm LT-LinUCB is given in Algorithm 1. It maintains the Gramian matrix $M_v$ and the moment vector $b_v$ of regressand by regressors to store the information for $w_v$. At each round $t$, the learner first computes the confidence ellipsoid for $w_v$ based on the current information (line 4) (see the following lemma).

**Lemma 1.** *Given $\{(A_t, y_t)\}_{t=1}^\infty$ with $A_t \in \{0,1\}^N$ and $y_t \in \{0,1\}$ as a Bernoulli random variable with $\mathbb{E}[y_t \mid A_1, y_1, \ldots, A_{t-1}, y_{t-1}, A_t] = A_t^\top w_v$, let $M_t = I + \sum_{s=1}^t A_s A_s^\top$ and $\hat{w}_t = M_t^{-1}\left(\sum_{s=1}^t A_s y_s\right)$ be the linear regression estimator. Then with probability at least $1 - \delta$, for all $t \geq 1$, it holds that $w_v$ lies in the confidence set*

$$\tilde{\mathcal{C}}_t := \left\{ w' \in [0,1]^N : \|w' - \hat{w}_t\|_{M_t} \leq \sqrt{N \log(1 + tN) + 2\log\frac{1}{\delta}} + \sqrt{N} \right\}.$$

---

**Algorithm 1** LT-LinUCB

1: **Input:** Graph $G = (V, E)$; seed set cardinality $K$; exploration parameter $\rho_{t,v} > 0$ for any $t, v$; offline oracle PairOracle
2: **Initialize:** $M_{0,v} \leftarrow I \in \mathbb{R}^{|N(v)| \times |N(v)|}, b_{0,v} \leftarrow 0 \in \mathbb{R}^{|N(v)| \times 1}, \hat{w}_{0,v} \leftarrow 0 \in \mathbb{R}^{|N(v)| \times 1}$ for any node $v \in V$
3: **for** $t = 1, 2, 3, \ldots$ **do**
4:     Compute the confidence ellipsoid $\mathcal{C}_{t,v} = \left\{ w'_v \in [0,1]^{|N(v)| \times 1} : \|w'_v - \hat{w}_{t-1,v}\|_{M_{t-1,v}} \leq \rho_{t,v} \right\}$
    for any node $v \in V$
5:     Compute the pair $(S_t, w_t)$ by PairOracle with confidence set $\mathcal{C}_t = \{\mathcal{C}_{t,v}\}_{v \in V}$ and seed set cardinality $K$
6:     Select the seed set $S_t$ and observe the feedback
7:     // Update
8:     **for** node $v \in V$ **do**
9:         Initialize $A_{t,v} \leftarrow 0 \in \mathbb{R}^{|N(v)| \times 1}, y_{t,v} \leftarrow 0 \in \mathbb{R}$
10:        Uniformly randomly choose $\tau \in \{\tau' : \tau_{t,1}(v) \leq \tau' \leq \tau_{t,2}(v) - 1\}$
11:        **if** $v$ is influenced and $\tau = \tau_{t,2}(v) - 1$ **then**
12:           $A_{t,v} = \chi(E_{t,\tau}(v)), \quad y_{t,v} = 1$
13:        **else if** $\tau = \tau_1(v), \ldots, \tau_2(v) - 2$ or $\tau = \tau_2(v) - 1$ but $v$ is not influenced **then**
14:           $A_{t,v} = \chi(E_{t,\tau}(v)), \quad y_{t,v} = 0$
15:        **end if**
16:        $M_{t,v} \leftarrow M_{t-1,v} + A_{t,v} A_{t,v}^\top, \quad b_{t,v} \leftarrow b_{t-1,v} + y_{t,v} A_{t,v}, \quad \hat{w}_{t,v} = M_{t,v}^{-1} b_{t,v}$
17:     **end for**
18: **end for**

---

This lemma is a direct corollary of [1, Theorem 2] for the concentration property of the weight vector $w_v$. Thus when $\rho_{t,v} \geq \sqrt{|N(v)| \log(1 + t|N(v)|) + 2\log\frac{1}{\delta}} + \sqrt{|N(v)|}$, the true weight vector $w_v$ lies in the confidence set $\mathcal{C}_{t,v}$ (line 4) for any $t$ with probability at least $1 - \delta$.

Given the confidence set $\mathcal{C}_v$ for $w_v$, the algorithm expects to select the seed set by solving the *weight-constrained influence maximization* (WCIM) problem

$$\text{argmax}_{(S,w'):S \in \mathcal{A}, w' \in \mathcal{C}} \, r(S, w'). \tag{5}$$

This (offline) optimization problem turns out to be highly nontrivial. Since we want to focus more on the online learning solution, we defer the full discussion on the offline optimization, including its general difficulty and our proposed approximate algorithms for certain graph classes such as directed acyclic graphs to Appendix B.

Suppose its best solution is $(S_{\mathcal{C}}^{\text{POpt}}, w_{\mathcal{C}}^{\text{POpt}})$ where 'P' stands for 'pair'. Let `PairOracle` be an offline oracle to solve the optimization problem. We say `PairOracle` is an $(\alpha, \beta)$-approximation oracle if $\mathbb{P}\left(r(S', w') \geq \alpha \cdot r(S_{\mathcal{C}}^{\text{POpt}}, w_{\mathcal{C}}^{\text{POpt}})\right) \geq \beta$ where $(S', w')$ is an output by the oracle when the confidence set is $\mathcal{C}$. Then the algorithm runs with the seed set output by the `PairOracle` and the confidence set $\mathcal{C}_t = \{\mathcal{C}_{t,v}\}_{v \in V}$ (line 5).

After observing the diffusion process (line 6), For each node $v$ who has active in-neighbors, we randomly choose its active in-neighbors at time step $\tau_1(v), \ldots, \tau_2(v) - 1$ to update (line 10). Specifically, if $v$ is influenced and $\tau = \tau_2(v) - 1$, then it means that the set of active in-neighbors at time step $\tau$ succeeds to activate $v$, thus we use $(\chi(E_{t,\tau}(v)), 1)$ to update (line 12); if $\tau = \tau_1(v), \ldots, \tau_2(v) - 2$ or $\tau = \tau_2(v) - 1$ but node $v$ is not influenced, it means that the set of active in-neighbors at $\tau$ fail to activate node $v$, thus we use $(\chi(E_{t,\tau}(v)), 0)$ to update (line 14). These updates are consistent with the distilled observations we get for nodes who have active in-neighbors. For node $v$ who has no active in-neighbors, we have no obervation on $w_v$ and not update on it since the set $\{\tau' : \tau_1(v) \leq \tau' \leq \tau_2(v) - 1\}$ is an empty set in this case.

For example in Figure 1, node $v_7$ has active in-neighbors from $\tau_1(v_7) = 1$ and is influenced at $\tau_2(v_7) = 3$. The `LT-LinUCB` will uniformly randomly choose $\tau \in \{1, 2\}$ (line 10). It updates $(A_{v_7} = \chi(E_1(v_7)), y_{v_7} = 0)$ if $\tau = 1$ (line 14) and $(A_{v_7} = \chi(E_2(v_7)), y_{v_7} = 1)$ otherwise (line 12). For nodes $v_1, v_2, v_3$, they all have $\tau_1 = 0$ and $\tau_2 = 1$. Thus for these three nodes, the algorithm chooses $\tau = 0$ (line 10) and updates $(A_v = \chi(E_0(v)), y_v = 1)$ (line 12). Node $v_4$ has active in-neighbors from $\tau_1(v_4) = 1$ but is not influenced finally, the algorithm will randomly choose $\tau \in \{1, 2 \ldots, D\}$ and update $(A_{v_4} = \chi(E_\tau(v_4)), y_{v_4} = 0)$ (line 14). Node $v_6$ has no active in-neighbors, so we have no observation for its weight vector and will not update on it.

## 3.1 Regret Analysis

We now provide the group observation modulated (GOM) bounded smoothness property for LT model, an important relationship of the influence spreads under two weight vectors. It plays a crucial role in the regret analysis and states that the difference of the influence spread $r(S, w)$ under two weight vectors can be bounded in terms of the weight differences of the distilled observed edge sets under one weight vector. It is conceptually similar to the triggering probability modulate (TPM) bounded smoothness condition under the IC model with edge-level feedback [47], but its derivation and usage are quite different. For the seed set $S$, define the set of all nodes related to a node $v$, $V_{S,v}$, to be the set of nodes that are on any path from $S$ to $v$ in graph $G$.

**Theorem 1.** *(GOM bounded smoothness) For any two weight vectors $w, w' \in [0, 1]^m$ with $\sum_{u \in N(v)} w(e_{u,v}) \leq 1$, the difference of their influence spread for any seed set $S$ can be bounded as*

$$|r(S, w') - r(S, w)| \leq \mathbb{E}\left[\sum_{v \in V \setminus S} \sum_{u \in V_{S,v}} \sum_{\tau = \tau_1(u)}^{\tau_2(u)-1} \left|\sum_{e \in E_\tau(u)} (w'(e) - w(e))\right|\right], \quad (6)$$

*where the definitions of $\tau_1(u), \tau_2(u)$ and $E_\tau(u)$ are all under weight vector $w$, and the expectation is taken over the randomness of the thresholds on nodes.*

This theorem connects the reward difference with weight differences on the distilled observations, which are also the information used to update the algorithm (line 7-17). It links the effective observations, updates of the algorithm and the regret analysis. The proof needs to deal with intricate dependency among activation events, and is put in Appendix A.1. due to the space constraint.

For seed set $S \in \mathcal{A}$ and node $u \in V \setminus S$, define $N_{S,u} := \sum_{v \in V \setminus S} \mathbb{1}\{u \in V_{S,v}\} \leq n - K$ to be the number of nodes that $u$ is relevant to. Then for the vector $N_S = (N_{S,u})_{u \in V}$, define the upper bound of its $L^2$-norm over all feasible seed sets

$$\gamma(G) := \max_{S \in \mathcal{A}} \sqrt{\sum_{u \in V} N_{S,u}^2} \leq (n - K)\sqrt{n} = O(n^{3/2}),$$

---
**Algorithm 2** `OIM-ETC`
---
1: **Input:** $G = (V, E)$, seed size $K$, exploration budget $k$, time horizon $T$, offline oracle `Oracle`
2: **for** $s \in [k], u \in V$ **do**
3:      Choose $\{u\}$ as the seed set
4:      $X_s(e_{u,v}) := \mathbb{1}\{v \text{ is activated}\}$ for any $v \in N^{\text{out}}(u)$
5: **end for**
6: Compute $\hat{w}(e) := \frac{1}{k} \sum_{s=1}^{k} X_s(e)$ for any $e \in E$
7: $\hat{S} = \text{Oracle}(\hat{w})$
8: **for** the remaining $T - nk$ rounds **do**
9:      Choose $\hat{S}$ as the seed set
10: **end for**
---

which is a constant related to the graph. Then we have the following regret bound.

**Theorem 2.** *Suppose the* `LT-LinUCB` *runs with an* $(\alpha, \beta)$*-approximation* `PairOracle` *and parameter* $\rho_{t,v} = \rho_t = \sqrt{n \log(1 + tn) + 2 \log \frac{1}{\delta}} + \sqrt{n}$ *for any node* $v \in V$. *Then the* $\alpha\beta$*-scaled regret satisfies*

$$R(T) \leq 2\rho_T \gamma(G) D \sqrt{mnT \log(1 + T) / \log(1 + n)} + n\delta \cdot T(n - k). \tag{7}$$

*When* $\delta = 1/(n\sqrt{T})$, $R(T) \leq C \cdot \gamma(G) \, Dn\sqrt{mT} \log(T)$ *for some universal constant* $C$.

Due to space limits, the proof and the detailed discussions, as well as the values of $\gamma(G)$, are put in Appendix A.

## 4   The Explore-then-Commit Algorithm

This section presents the explore-then-commit (ETC) algorithm for OIM. Though simple, it is efficient and model independent, applying to both LT and IC model with less requirement on feedback and offline computation.

Recall that under LT model, a node $v$ is activated if the sum of weights from active in-neighbors exceeds the threshold $\theta_v$, which is uniformly drawn from $[0, 1]$. Since the feedback is node-level, if the activated node $v$ has more than one active in-neighbors, then we can only observe the group influence effect of her active in-neighbors instead of each single in-neighbor. A simple way to overcome this limitation and manage to observe directly the single weight $w(e_{u,v})$ is to select a single seed $\{u\}$ and take only the first step influence as feedback, which formulates our `OIM-ETC` algorithm (Algorithm 2), representing the ETC algorithm of the OIM problem.

Our `OIM-ETC` takes the exploration budget $k$ as input parameter such that each node $u$ is selected as the (single) seed for $k$ rounds (line 3). For each round in which $u$ is the seed, each outgoing neighbor (shortened as out-neighbor) $v \in N^{\text{out}}(u)$ will be activated in the first step with probability $\mathbb{P}(w(e_{u,v}) > \theta_v) = w(e_{u,v})$ since the threshold $\theta_v$ is independently uniformly drawn from $[0, 1]$. Thus the first-step node-level feedback is actually edge-level feedback and we can observe the independent edges from the first-step feedback (line 4). Since each node is selected $k$ times, we have $k$ observations of Bernoulli random variables with expectation $w(e_{u,v})$ in this *exploration* phase. Then we take the empirical estimate $\hat{w}(e)$ for each $w(e)$ (line 6) after the exploration and run with the seed set output by the offline `Oracle` (line 7) for the remaining $T - nk$ *exploitation* rounds (line 9). We assume the offline `Oracle` is $(\alpha, \beta)$-approximation.

Since it only needs the first step of the diffusion process and calls only once of the usual IM oracle, it is efficient and has less requirement. By selecting reasonable $k$, we can derive good regret bounds. Before that we need two definitions.

**Definition 1.** *(Bad seed set) A seed set* $S$ *is bad if* $r(S, w) < \alpha \cdot \text{Opt}_w$. *The set of bad seed sets is* $\mathcal{S}_B := \{S \mid r(S, w) < \alpha \cdot \text{Opt}_w\}$.

**Definition 2.** *(Gaps of bad seed sets) For a bad seed set* $S \in \mathcal{S}_B$, *its gap is defined as* $\Delta_S := \alpha \cdot \text{Opt}_w - r(S, w)$. *The maximum and minimum gap are defined as*

$$\Delta_{\max} := \alpha \cdot \text{Opt}_w - \min\{r(S, w) \mid S \in \mathcal{S}_B\}, \tag{8}$$

$$\Delta_{\min} := \alpha \cdot \text{Opt}_w - \max\{r(S, w) \mid S \in \mathcal{S}_B\}. \tag{9}$$

**Theorem 3.** *When $k = \max\left\{1, \frac{2m^2 n^2}{\Delta_{\min}^2} \ln\left(\frac{T\Delta_{\min}^2}{mn^3}\right)\right\}$, the $\alpha\beta$-scaled regret bound of our* `OIM-ETC` *algorithm over $T$ rounds satisfies*

$$R(T) \le \min\left\{T\Delta_{\max}, n\Delta_{\max} + \frac{2m^2 n^3 \Delta_{\max}}{\Delta_{\min}^2}\left(1 + \max\left\{0, \ln\left(\frac{T\Delta_{\min}^2}{mn^3}\right)\right\}\right)\right\}$$

$$= O\left(\frac{m^2 n^3 \Delta_{\max}}{\Delta_{\min}^2} \ln(T)\right). \tag{10}$$

*When $k = 3.9(m^2 T/n)^{2/3}$, the $\alpha\beta$-scaled regret bound of* `OIM-ETC` *algorithm over $T$ rounds satisfies*

$$R(T) \le 3.9(mn)^{4/3} T^{2/3} + 1 = O\left((mn)^{4/3} T^{2/3}\right). \tag{11}$$

The proof of the problem-dependent bound follows routine ideas of ETC algorithms but the proof of the problem-independent bound is new. The proofs and discussions are put in Appendix C.

## 5 Conclusion

In this paper, we formulate the problem of OIM under LT model with node-level feedback and design how to distill effective information from observations. We prove a novel GOM bounded smoothness property for the spread function, which relates the limited observations, algorithm updates and the regret analysis. We propose `LT-LinUCB` algorithm, provide rigorous theoretical analysis and show a competitive regret bound of $O(\mathrm{poly}(m)\sqrt{T}\ln(T))$. Our `LT-LinUCB` is the first algorithm for LT model with such regret order. Besides, we design `OIM-ETC` algorithm with theoretical analysis on its distribution-dependent and distribution-independent regret bounds. The algorithm is efficient, applies to both LT and IC models, and has less requirements on feedback and offline computation.

In studying the OIM with LT model, we encounter an optimization problem of weight-constrained influence maximization (WCIM). Reconsidering an (offline) optimization problem by relaxing some fixed parameter to elements of a convex set is expected to be common in online learning. So we believe this problem could have independent interest. Also the OIM problem under IC model with node-level feedback is an interesting future work. Our regret analysis goes through thanks to the linearity of the LT model. But the local triggering is nonlinear for IC model, and thus we expect more challenges in the design and analysis of IC model with node-level feedback. Applying Thompson sampling to influence maximization is also an interesting future direction, but it could also be challenging, since it may not work well with offline approximation oracles as pointed out in [48].

## Acknowledgement

This work is sponsored by Shanghai Sailing Program. We thank Chihao Zhang for valuable discussions.

## Broader Impact

Spread happens everywhere, no matter when a company wants to advertise their products, a group wants to seek attention, or even virus like COVID-19 walks between places. The modelling of a spread is always based on a directed graph and uses some diffusion assumptions. The linear threshold (LT) model, considered in our paper, is one of the most popular models. The influence maximization is the problem to pursue the widest spread, while the online version is to adaptively achieve this goal through an interactive manner with no knowledge of the parameters.

Compared with application research, theoretical analysis will provide analysis and guarantees for the designed algorithms. The theoretical work usually include extreme cases to make sure the algorithm does have a good property, while in application research the possible experiments are always limited and usually do not reflect corner cases. The latter might be an issue if we want to transfer the model to some unseen scenes, e.g. automatic drive.

Also among our derivations, we reveal a class of optimization problems (i.e. WCIM), which we expect to happen a lot if the community wants to solve some (offline) problem without the knowledge of the parameters using an online manner. This class, as a theoretical problem, is highly non-trivial and can introduce new problems to the optimization area. The hardness of this arising problem also reflects the difficulty of transforming offline or heuristic solutions to online or automatic ones, like the AutoML direction and the topics of automatically adjusting hyper-parameters.

## Footnotes

[2]One may think of the node-level feedback as knowing only the set of nodes activated by the end of the diffusion process. We refer to this as (partial) node-level feedback and ours as (full) node-level feedback. This naming comes from [35].

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
