[Supplementary Material]

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

# A  Analysis and Discussions of `LT-LinUCB`

## A.1  Proof of Theorem 1

Let $r_S^v(w)$ be the probability that node $v$ will be influenced under the weight vector $w$ when the seed set is $S$. Then

$$|r(S, w') - r(S, w)|$$
$$\leq \sum_{v \in V \setminus S} |r_S^v(w') - r_S^v(w)|$$
$$= \sum_{v \in V \setminus S} \mathbb{E}_{\theta \sim (\mathcal{U}[0,1])^n} \left[ \mathbb{1}\{v \text{ is influenced under } w', \theta\} \neq \mathbb{1}\{v \text{ is influenced under } w, \theta\} \right],$$

where we use $\mathcal{U}[0,1]$ to denote the uniform distribution on the interval $[0,1]$. The reason that the activation of $v$ is different under $w$ and $w'$ must be that during the propagation from $S$ to $v$, at some step $\tau$ and some node $u \in V_{S,v}$, the activation of $u$ is different. We enumerate $u \in V_{S,v}$ and enumerate $\tau$ from 1 to $D$ to bound the above probability. Recall that $D$ is the diameter. Henceforth in this section, parameters $w$, $w'$, $S$, and $v$ are all fixed. All the randomness comes from $\theta \sim (\mathcal{U}[0,1])^n$, and once $\theta$ is determined, the diffusion process is determined. Thus, we could assume that every event is a subset of $[0,1]^n$. Define the following event, given the seed set $S$ and target node $v$:

$$\mathcal{E}_0 = \{\theta \mid \mathbb{1}\{v \text{ is influenced under } w', \theta\} \neq \mathbb{1}\{v \text{ is influenced under } w, \theta\}\}.$$

Thus

$$|r(S, w') - r(S, w)| \leq \sum_{v \in V \setminus S} \Pr_{\theta \sim (\mathcal{U}[0,1])^n} \{\mathcal{E}_0\}. \tag{12}$$

Let $\Phi(w, \theta) = (S_0 = S, S_1, \ldots, S_D)$ be the sequence of activation sets given weight factor $w$ and threshold factor $\theta$. Let $\Phi_i(w, \theta) = S_i$ be the set of nodes activated by time step $i$. For every node $u \in V_{S,v}$, we define the event that $u$ is the first node that has different activation under $w$ and $w'$.

$$\mathcal{E}_1(u) = \{\theta \mid \exists \tau \in [D], \forall \tau' < \tau, \Phi_{\tau'}(w, \theta) = \Phi_{\tau'}(w', \theta),$$
$$u \in (\Phi_\tau(w, \theta) \setminus \Phi_\tau(w', \theta)) \cup (\Phi_\tau(w', \theta) \setminus \Phi_\tau(w, \theta))\}.$$

It is clear that

$$\mathcal{E}_0 \subseteq \bigcup_{u \in V_{S,v}} \mathcal{E}_1(u). \tag{13}$$

Note that for each node $u \in V_{S,v}$, $u$ may be activated at different time steps from different paths, or not activated at all. Thus, the fact that $u$ is not activated at one time step may have implications on $u$'s activations at other time steps, and thus we need to carefully classify the activation of $u$ in order to bound the probability of $\mathcal{E}_1(u)$. Define the following events for each $\tau \in [D]$:

$$\mathcal{E}_{2,0}(u, \tau) = \{\theta \mid \forall \tau' < \tau, \Phi_{\tau'}(w, \theta) = \Phi_{\tau'}(w', \theta), u \notin \Phi_{\tau-1}(w, \theta)\},$$
$$\mathcal{E}_{2,1}(u, \tau) = \{\theta \mid \forall \tau' < \tau, \Phi_{\tau'}(w, \theta) = \Phi_{\tau'}(w', \theta), u \in \Phi_\tau(w, \theta) \setminus \Phi_\tau(w', \theta)\},$$
$$\mathcal{E}_{2,2}(u, \tau) = \{\theta \mid \forall \tau' < \tau, \Phi_{\tau'}(w, \theta) = \Phi_{\tau'}(w', \theta), u \in \Phi_\tau(w', \theta) \setminus \Phi_\tau(w, \theta)\},$$
$$\mathcal{E}_{3,1}(u, \tau) = \{\theta \mid u \in \Phi_\tau(w, \theta) \setminus \Phi_\tau(w', \theta)\},$$
$$\mathcal{E}_{3,2}(u, \tau) = \{\theta \mid u \in \Phi_\tau(w', \theta) \setminus \Phi_\tau(w, \theta)\}.$$

Note that all the events $\mathcal{E}_{2,1}(u, \tau), \mathcal{E}_{2,2}(u, \tau)$ for $\tau \in [D]$ are mutually exclusive. Therefore,

$$\Pr_{\theta \sim (\mathcal{U}[0,1])^n} \{\mathcal{E}_1(u)\} = \sum_{\tau=1}^{D} \Pr_{\theta \sim (\mathcal{U}[0,1])^n} \{\mathcal{E}_{2,1}(u, \tau)\} + \sum_{\tau=1}^{D} \Pr_{\theta \sim (\mathcal{U}[0,1])^n} \{\mathcal{E}_{2,2}(u, \tau)\}. \tag{14}$$

We first bound $\Pr_{\theta \sim (\mathcal{U}[0,1])^n} \{\mathcal{E}_{2,1}(u, \tau)\}$. Now fix all entries of $\theta$ vector except $\theta_u$, denoted as $\theta_{-u}$, and the corresponding subevent of $\mathcal{E}_{2,1}(u, \tau)$ is defined as $\mathcal{E}_{2,1}(u, \tau, \theta_{-u}) \subseteq \mathcal{E}_{2,1}(u, \tau)$. Similarly $\mathcal{E}_{2,0}(u, \tau, \theta_{-u}) \subseteq \mathcal{E}_{2,0}(u, \tau)$ and $\mathcal{E}_{3,1}(u, \tau, \theta_{-u}) \subseteq \mathcal{E}_{3,1}(u, \tau)$ are defined. Also $\mathcal{E}_{2,1}(u, \tau, \theta_{-u}) = \mathcal{E}_{2,0}(u, \tau, \theta_{-u}) \cap \mathcal{E}_{3,1}(u, \tau, \theta_{-u})$.

Note that $\mathcal{E}_{2,1}(u,\tau) = \mathcal{E}_{2,0}(u,\tau) \cap \mathcal{E}_{3,1}(u,\tau)$, and $\mathcal{E}_{2,2}(u,\tau) = \mathcal{E}_{2,0}(u,\tau) \cap \mathcal{E}_{3,2}(u,\tau)$. Thus

$$\Pr_{\theta \sim (\mathcal{U}[0,1])^n}\{\mathcal{E}_{2,1}(u,\tau)\} = \Pr_{\theta \sim (\mathcal{U}[0,1])^n}\{\mathcal{E}_{2,0}(u,\tau)\} \cdot \Pr_{\theta \sim (\mathcal{U}[0,1])^n}\{\mathcal{E}_{3,1}(u,\tau) \mid \mathcal{E}_{2,0}(u,\tau)\}. \quad (15)$$

Then

$$\Pr_{\theta_u \sim \mathcal{U}[0,1]}\{\mathcal{E}_{2,1}(u,\tau,\theta_{-u})\} = \Pr_{\theta_u \sim \mathcal{U}[0,1]}\{\mathcal{E}_{2,0}(u,\tau,\theta_{-u})\} \cdot \Pr_{\theta_u \sim \mathcal{U}[0,1]}\{\mathcal{E}_{3,1}(u,\tau,\theta_{-u}) \mid \mathcal{E}_{2,0}(u,\tau,\theta_{-u})\}. \quad (16)$$

Symmetric equations also hold for $\mathcal{E}_{2,2}(u,\tau,\theta_{-u})$.

In the event $\mathcal{E}_{2,0}(u,\tau,\theta_{-u})$, all entries in $\theta$ vector is fixed except for $\theta_u$. It is easy to check that if $(\theta_{-u},\theta_u) \in \mathcal{E}_{2,0}(u,\tau,\theta_{-u})$, then for all $\theta'_u \geq \theta_u$, $(\theta_{-u},\theta'_u) \in \mathcal{E}_{2,0}(u,\tau,\theta_{-u})$. This is because the $\theta_{-u}$ is fixed, so the activations of all nodes other than $u$ have the same conditions, while for $u$ it is even harder to activate $u$ with larger $\theta_u$. Therefore, in $\mathcal{E}_{2,0}(u,\tau,\theta_{-u})$, the entry on $\theta_u$ must be an interval from some lowest value to 1. Let $\theta_{u,2,0}(\tau,\theta_{-u})$ be the left point of this interval. That is $\mathcal{E}_{2,0}(u,\tau,\theta_{-u}) = \{(\theta_{-u},\theta_u) \mid \theta_u > \theta_{u,2,0}(\tau,\theta_{-u})\}$. Then we have

$$\Pr_{\theta_u \sim \mathcal{U}[0,1]}\{\mathcal{E}_{2,0}(u,\tau,\theta_{-u})\} = 1 - \theta_{u,2,0}(\tau,\theta_{-u}). \quad (17)$$

For now, let's first assume that $\mathcal{E}_{2,0}(u,\tau,\theta_{-u}) \neq \emptyset$, that is, $\theta_{u,2,0}(\tau,\theta_{-u}) < 1$. In the event $\mathcal{E}_{2,0}(u,\tau,\theta_{-u})$, we know that the set of activated nodes until $\tau - 1$ are the same under both $w$ and $w'$ and $u$ is not activated by time $\tau - 1$, and since $\theta_{-u}$ is fixed, the set of activated nodes by time $\tau - 1$ are all fixed. We denote the set of nodes activated by time step $i$ under event $\mathcal{E}_{2,0}(u,\tau,\theta_{-u})$ as $\Phi_i(\mathcal{E}_{2,0}(u,\tau,\theta_{-u}))$.

Now conditioned on the event $\mathcal{E}_{2,0}(u,\tau,\theta_{-u})$, we consider event $\mathcal{E}_{3,1}(u,\tau,\theta_{-u}) \cup \mathcal{E}_{3,2}(u,\tau,\theta_{-u})$. This means that conditioned on $\theta_u > \theta_{u,2,0}(\tau,\theta_{-u})$ and a fixed activated set $\Phi_{\tau-1}(\mathcal{E}_{2,0}(u,\tau,\theta_{-u}))$ by time $\tau - 1$, $u$ is activated at step $\tau$ under one of $w$ and $w'$ but not both. According to the information diffsuion under the LT model, this means either the following inequality holds,

$$\sum_{u' \in \Phi_{\tau-1}(\mathcal{E}_{2,0}(u,\tau,\theta_{-u})) \cap N(u)} w'(e_{u',u}) < \theta_u \leq \sum_{u' \in \Phi_{\tau-1}(\mathcal{E}_{2,0}(u,\tau,\theta_{-u})) \cap N(u)} w(e_{u',u}).$$

or the following holds

$$\sum_{u' \in \Phi_{\tau-1}(\mathcal{E}_{2,0}(u,\tau,\theta_{-u})) \cap N(u)} w(e_{u',u}) < \theta_u \leq \sum_{u' \in \Phi_{\tau-1}(\mathcal{E}_{2,0}(u,\tau,\theta_{-u})) \cap N(u)} w'(e_{u',u}).$$

This in turn implies that

$$\Pr_{\theta_u \sim \mathcal{U}[0,1]}\{\mathcal{E}_{3,1}(u,\tau,\theta_{-u}) \cup \mathcal{E}_{3,2}(u,\tau,\theta_{-u}) \mid \mathcal{E}_{2,0}(u,\tau,\theta_{-u})\}$$

$$= \frac{\left|\sum_{u' \in \Phi_{\tau-1}(\mathcal{E}_{2,0}(u,\tau,\theta_{-u})) \cap N(u)} w(e_{u',u}) - \sum_{u' \in \Phi_{\tau-1}(\mathcal{E}_{2,0}(u,\tau,\theta_{-u})) \cap N(u)} w'(e_{u',u})\right|}{1 - \theta_{u,2,0}(\tau,\theta_{-u})}.$$

Plugging the above equality and Eq.(17) into Eq.(16), and use the fact that $\mathcal{E}_{3,1}(u,\tau,\theta_{-u})$ and $\mathcal{E}_{3,2}(u,\tau,\theta_{-u})$ are mutually exclusive, we have

$$\Pr_{\theta_u \sim \mathcal{U}[0,1]}\{\mathcal{E}_{2,1}(u,\tau,\theta_{-u}) \cup \mathcal{E}_{2,2}(u,\tau,\theta_{-u})\}$$

$$= \left|\sum_{u' \in \Phi_{\tau-1}(\mathcal{E}_{2,0}(u,\tau,\theta_{-u})) \cap N(u)} (w(e_{u',u}) - w'(e_{u',u}))\right|. \quad (18)$$

Note that when $\mathcal{E}_{2,0}(u,\tau,\theta_{-u}) = \emptyset$, both the LHS and the RHS of the above equality is zero, so this equality holds in general.

We now need to relax event $\mathcal{E}_{2,0}(u,\tau,\theta_{-u})$, since it depends on both $w$ and $w'$. We define a new event to detach it from $w'$,

$$\mathcal{E}_{4,0}(u,\tau,\theta_{-u}) = \{\theta = (\theta_{-u},\theta_u) \mid u \notin \Phi_{\tau-1}(w,\theta)\}.$$

It is clear that $\mathcal{E}_{2,0}(u,\tau,\theta_{-u}) \subseteq \mathcal{E}_{4,0}(u,\tau,\theta_{-u})$. Moreover, when $\mathcal{E}_{2,0}(u,\tau,\theta_{-u}) \neq \emptyset$, we see that both events $\mathcal{E}_{2,0}(u,\tau,\theta_{-u})$ and $\mathcal{E}_{4,0}(u,\tau,\theta_{-u})$ have fixed $\theta_{-u}$ and dictate that $u$ is not activated by time $\tau - 1$ under $w$. This implies that they have the same set of nodes activated by time step $i$ for $i \leq \tau - 1$. Denote $\Phi_i(\mathcal{E}_{4,0}(u,\tau,\theta_{-u}))$ be this set. The above means that for all $i \leq \tau - 1$, $\Phi_i(\mathcal{E}_{2,0}(u,\tau,\theta_{-u})) = \Phi_i(\mathcal{E}_{4,0}(u,\tau,\theta_{-u}))$. Therefore, we can relax Eq.(18) to get the following.

$$\Pr_{\theta_u \sim \mathcal{U}[0,1]}\{\mathcal{E}_{2,1}(u,\tau,\theta_{-u}) \cup \mathcal{E}_{2,2}(u,\tau,\theta_{-u})\}$$

$$\leq \left| \sum_{u' \in \Phi_{\tau-1}(\mathcal{E}_{4,0}(u,\tau,\theta_{-u})) \cap N(u)} (w(e_{u',u}) - w'(e_{u',u})) \right|.$$

Note that when $\mathcal{E}_{2,0}(u,\tau,\theta_{-u}) = \emptyset$, the LHS of above is zero, so the inequality still holds.

Combining the above with Eq.(14), we have

$$\Pr_{\theta \sim (\mathcal{U}[0,1])^n}\{\mathcal{E}_1(u)\}$$

$$= \int_{\theta_{-u} \in [0,1]^{n-1}} \sum_{\tau=1}^{D} \Pr_{\theta_u \sim \mathcal{U}[0,1]}\{\mathcal{E}_{2,1}(u,\tau,\theta_{-u}) \cup \mathcal{E}_{2,2}(u,\tau,\theta_{-u})\}\, d\theta_{-u}$$

$$= \sum_{\tau=1}^{D} \int_{\theta_{-u} \in [0,1]^{n-1}} \Pr_{\theta_u \sim \mathcal{U}[0,1]}\{\mathcal{E}_{2,1}(u,\tau,\theta_{-u}) \cup \mathcal{E}_{2,2}(u,\tau,\theta_{-u})\}\, d\theta_{-u}$$

$$\leq \sum_{\tau=1}^{D} \int_{\theta_{-u} \in [0,1]^{n-1}} \left| \sum_{u' \in \Phi_{\tau-1}(\mathcal{E}_{4,0}(u,\tau,\theta_{-u})) \cap N(u)} (w(e_{u',u}) - w'(e_{u',u})) \right| d\theta_{-u}$$

$$= \sum_{\tau=1}^{D} \mathbb{E}_{\theta_{-u} \sim (\mathcal{U}[0,1])^{n-1}} \left[ \left| \sum_{u' \in \Phi_{\tau-1}(\mathcal{E}_{4,0}(u,\tau,\theta_{-u})) \cap N(u)} (w(e_{u',u}) - w'(e_{u',u})) \right| \right].$$

Combining the above with Eq.(12) and Eq.(13), we have

$$|r(S,w') - r(S,w)|$$

$$\leq \sum_{v \in V \setminus S} \sum_{u \in V_{S,v}} \sum_{\tau=1}^{D} \mathbb{E}_{\theta_{-u} \sim (\mathcal{U}[0,1])^{n-1}} \left[ \left| \sum_{u' \in \Phi_{\tau-1}(\mathcal{E}_{4,0}(u,\tau,\theta_{-u})) \cap N(u)} (w(e_{u',u}) - w'(e_{u',u})) \right| \right]$$

$$= \mathbb{E}\left[ \sum_{v \in V \setminus S} \sum_{u \in V_{S,v}} \sum_{\tau=\tau_1(u)}^{\tau_2(u)-1} \left| \sum_{e \in E_\tau(u)} (w(e) - w'(e)) \right| \right],$$

where the last equality comes from the definition of $\tau_1(u), \tau_2(u)$ and $E_\tau(u)$ under weight vector $w$ and the expectation is taken over the randomness of the thresholds on nodes, specifically the value of $\tau_1(u), \tau_2(u)$ and $E_\tau(u)$ for each time step $\tau$. Thus we get the desired result.

## A.2  Proof of the Regret

The key Theorem 1 describes the difference of the influence spread under two weight vectors in terms of the (expected) weight differences of some edge sets, which coincides with the possible observations under LT model. So this theorem justifies why we distill the information and design the updates of the algorithm `LT-LinUCB` in this way. Next lemma further states the rationality explicitly. Recall that $w$ is the (unknown) true weight vector.

**Lemma 2.** *Let $S, w'$ be the seed set and the weight vector output at line 5 of* `PairOracle` *in a round $t$. Then for each fixed threshold $\theta \in [0,1]^n$,*

$$\sum_{\tau=\tau_1(u)}^{\tau_2(u)-1} \left| \sum_{e \in E_\tau(u)} (w'(e) - w(e)) \right| \leq D \cdot \mathbb{E}\left[ \left| A_u^\top (w'_u - w_u) \right| \right],$$

*where the definitions of $\tau_1(u), \tau_2(u)$ and $E_\tau(u)$ are defined under weight vector $w$, $A_u$ is the value of $A_{t,u}$ updated in lines 11–15 in round $t$, which is the distilled edge set chosen by* LT-LinUCB *to update for node $u$, and the expectation is taken over the randomness of $\tau$ (line 10) in determining $A_u$ when $u$ has active in-neighbors.*

*Proof.* Let $D_u = \tau_2(u) - \tau_1(u)$. If $u$ has active in-neighbors, then according to line 7-17 of the Algorithm 1, $A_u = \chi(E_\tau(u))$ where $\tau = \tau_1(u), \ldots, \tau_2(u) - 1$ with probability $1/D_u$ respectively. Thus,

$$\mathbb{E}\left[\left|A_u^\top (w'_u - w_u)\right|\right] = \frac{1}{D_u} \sum_{\tau=\tau_1(u)}^{\tau_2(u)-1} \left|\chi(E_\tau(u))^\top (w'_u - w_u)\right|$$

$$= \frac{1}{D_u} \left( \sum_{\tau=\tau_1(u)}^{\tau_2(u)-1} \left| \sum_{e \in E_\tau(u)} (w'(e) - w(e)) \right| \right) .$$

Since the diffusion process lasts for at most $D$ steps, it is straightforward that $D_u \leq D$, thus we get the inequality holds.

If $u$ has no active in-neighbors, then by definition the values of both LHS and RHS are 0, thus the inequality still holds. $\qquad\square$

Now we are ready to prove the regret bound.

*Proof of Theorem 2.* Define the failure event

$$\mathcal{F} = \left\{ \exists t \leq T, v \in V : \|w_v - \hat{w}_{t,v}\|_{M_{t,v}} > \rho_{t,v} \right\} \tag{19}$$

to represent the true weight vector $w_v$ does not lie in the confidence ellipsoid $\mathcal{C}_{t,v}$ for some round $t$ and node $v$. Then by Lemma 1, when $\rho_{t,v} = \rho_t = \sqrt{n \log(1 + tn) + 2\log\frac{1}{\delta}} + \sqrt{n}$, $\mathcal{F}^c$ holds with probability at least $1 - n\delta$. Next we bound the regret conditioned on the event $\mathcal{F}^c$.

Recall that PairOracle is an $(\alpha, \beta)$-approximation oracle, adopted in LT-LinUCB. Then the $(\alpha, \beta)$-scaled regret of round $t$ satisfies

$$\mathbb{E}\left[R_t\right] = \mathbb{E}\left[\alpha\beta \cdot \mathrm{Opt}_w - r(S_t, w)\right] \leq \mathbb{E}\left[\alpha\beta \cdot r(S_{\mathcal{C}_t}^{\mathrm{POpt}}, w_{\mathcal{C}_t}^{\mathrm{POpt}}) - r(S_t, w)\right]$$
$$\leq \mathbb{E}\left[r(S_t, w_t) - r(S_t, w)\right],$$

where the last inequality is by the property that PairOracle is $(\alpha, \beta)$-approximation, and the expectation is over the randomness of the oracle and the randomness in the influence spread.

Then by Theorem 1 and Lemma 2,

$$\mathbb{E}\left[r(S_t, w_t) - r(S_t, w)\right] \leq D \cdot \mathbb{E}\left[ \sum_{v \in V \setminus S_t} \sum_{u \in V_{S_t, v}} \left|A_{t,u}^\top (w_{t,u} - w_u)\right| \right]$$

$$\leq D \cdot \mathbb{E}\left[ \sum_{v \in V \setminus S_t} \sum_{u \in V_{S_t, v}} \|A_{t,u}\|_{M_{t,u}^{-1}} \|w_{t,u} - w_u\|_{M_{t,u}} \right]$$

$$\leq D \cdot \mathbb{E}\left[ \sum_{v \in V \setminus S_t} \sum_{u \in V_{S_t, v}} 2\rho_t \|A_{t,u}\|_{M_{t,u}^{-1}} \right],$$

since $w_{t,u}, w_u$ are both in the confidence set. Thus

$$R(T) = \mathbb{E}\left[ \sum_{t=1}^T R_t \right] \leq 2\rho_T D \cdot \mathbb{E}\left[ \sum_{t=1}^T \sum_{v \in V \setminus S_t} \sum_{u \in V_{S_t, v}} \|A_{t,u}\|_{M_{t,u}^{-1}} \right]$$

$$\leq 2\rho_T D \cdot \mathbb{E}\left[\sqrt{\left(\sum_{t=1}^{T}\sum_{u \in V} N_{S_t,u}^2\right)\left(\sum_{t=1}^{T}\sum_{u \in V} \|A_{t,u}\|_{M_{t,u}^{-1}}^2\right)}\right]$$

$$\leq 2\rho_T D \cdot \mathbb{E}\left[\sqrt{T}\gamma(G) \cdot \sqrt{\left(\sum_{t=1}^{T}\sum_{u \in V} \|A_{t,u}\|_{M_{t,u}^{-1}}^2\right)}\right]$$

where the second line is by Cauchy-Schwartz inequality.

Note that $M_{t,u} = M_{t-1,u} + A_{t,u}A_{t,u}^\top$ and

$$\det(M_{t,u}) = \det\left(M_{t-1,u} + A_{t,u}A_{t,u}^\top\right)$$
$$= \det\left(M_{t-1,u}^{1/2}\left(I + M_{t-1,u}^{-1/2}A_{t,u}A_{t,u}^\top M_{t-1,u}^{-1/2}\right)M_{t-1,u}^{1/2}\right)$$
$$= \det(M_{t-1,u})\det\left(I + M_{t-1,u}^{-1/2}A_{t,u}A_{t,u}^\top M_{t-1,u}^{-1/2}\right)$$
$$= \det(M_{t-1,u})\left(1 + \|A_{t,u}\|_{M_{t-1,u}^{-1}}^2\right)$$

where the last inequality holds because the determinant of a matrix is the product of its eigenvalues and the matrix $I + xx^\top$ has eigenvalues 1 and $1 + \|x\|_2^2$. And here $\left\|M_{t-1}^{-1/2}A_{t,u}\right\| = \|A_{t,u}\|_{M_{t-1}^{-1}}$. Then

$$\sum_{t=1}^{T}\sum_{u \in V}\|A_{t,u}\|_{M_{t,u}^{-1}}^2 \leq \sum_{t=1}^{T}\sum_{u \in V}\frac{n}{\log(1+n)}\cdot\log\left(1 + \|A_{t,u}\|_{M_{t,u}^{-1}}^2\right) \tag{20}$$

$$\leq \sum_{u \in V}\frac{n}{\log(1+n)}\log\frac{\det(M_{T,u})}{\det(I)}$$

$$\leq \sum_{u \in V}\frac{n\,|N(u)|}{\log(1+n)}\log(\text{trace}(M_{T,u})/\,|N(u)|) \tag{21}$$

$$\leq \sum_{u \in V}\frac{n\,|N(u)|}{\log(1+n)}\log\left(1 + \sum_{t=1}^{T}\|A_{t,u}\|_2^2 /\,|N(u)|\right)$$

$$\leq \sum_{u \in V}\frac{n\,|N(u)|}{\log(1+n)}\log(1+T)$$

$$= \frac{n}{\log(1+n)}\log(1+T)\cdot\sum_{u \in V}|N(u)|$$

$$= \frac{nm}{\log(1+n)}\log(1+T), \tag{22}$$

where (20) is by the inequality that $u \leq \frac{a}{\log(1+a)}\log(1+u)$ for $u \in [0,a]$ and $\|A_{t,u}\|_2^2 \leq |N(u)| \leq n$; (21) is by the inequality that $\det(M_{T,u}) \leq (\text{trace}(M_{T,u})/\,|N(u)|)^{|N(u)|}$ and (22) holds obviously since the sum of the number of in-neighbors of all nodes is just the number $m$ of edges in the graph.

Therefore the $\alpha\beta$-scaled regret satisfies

$$R(T) \leq 2\rho_T\gamma(G)D\sqrt{mnT\log(1+T)/\log(1+n)} + n\delta \cdot T(n-k)$$
$$\leq C \cdot \gamma(G)Dnm^{1/2}\sqrt{T}\log(T),$$

for some universal constant $C$. $\qquad\square$

### A.3 Discussions

**Comparisons of regret bounds** We compute the $\gamma(G)$ and $D$ for some special graphs and compare our regret bound with the IMLinUCB algorithm [49] and CUCB algorithm [47], where these two are under IC model and edge-level feedback. The results are listed in Table 1 where we use

the same examples as in [49, Figure 1]. For general graphs, our `LT-LinUCB` has regret bound $O(\gamma(G)Dnm^{1/2}\sqrt{T}\ln(T)) = O(n^{7/2}m^{1/2}\sqrt{T}\ln(T))$, the IMLinUCB algorithm has regret bound (in the tabular case) $O(C_G m\sqrt{T}\ln(T)) = O(nm^{3/2}\sqrt{T}\ln(T))$ and the CUCB algorithm has regret bound $O(B_G\sqrt{mK'T\ln(T)}) = O(mn\sqrt{T\ln(T)})$. So ours is at most $O(n^{5/2}/m)$ worse than IMLinUCB and $O(n^{5/2}\sqrt{\ln(T)}/\sqrt{m})$ worse than CUCB. Note that the freedom degree of LT model is $O(n)$ as there are only $n$ random variable $((\theta_v)_{v\in V})$ while the freedom degree of IC model is $O(m)$. Also we assume only node-level feedback is observed while edge-level feedback can be observed in their work on IC model.

| Graphs | $D$ | $\gamma(G)$ | `LT-LinUCB` (ours) | IMLinUCB | CUCB |
|---|---|---|---|---|---|
| bar graph | $O(1)$ | $O(\sqrt{K})$ | $O(n^{3/2}\sqrt{KT}\ln(T))$ | $O(n\sqrt{KT}\ln(T))$ | $O(\sqrt{nKT\ln(T)})$ |
| star graph | $O(1)$ | $O(n\sqrt{K})$ | $O(n^{5/2}\sqrt{KT}\ln(T))$ | $O(n^2\sqrt{KT}\ln(T))$ | $O(n^2\sqrt{T\ln(T)})$ |
| ray graph | $O(\sqrt{n})$ | $O(n^{5/4}\sqrt{K})$ | $O(n^{13/4}\sqrt{KT}\ln(T))$ | $O(n^{9/4}\sqrt{KT}\ln(T))$ | $O(n^2\sqrt{T\ln(T)})$ |
| tree graph | $O(\log n)$ | $O(n^{3/2})$ | $O(n^3\log n\sqrt{T}\ln(T))$ | $O(n^{5/2}\sqrt{T}\ln(T))$ | $O(n^2\sqrt{T\ln(T)})$ |
| grid graph | $O(n)$ | $O(n^{3/2})$ | $O(n^4\sqrt{T}\ln(T))$ | $O(n^{5/2}\sqrt{T}\ln(T))$ | $O(n^2\sqrt{T\ln(T)})$ |
| complete graph | $O(n)$ | $O(n^{3/2})$ | $O(n^{9/2}\sqrt{T}\ln(T))$ | $O(n^4\sqrt{T}\ln(T))$ | $O(n^3\sqrt{T\ln(T)})$ |

Table 1: The values of $\gamma(G)$, $D$ and regret bound comparisons of `LT-LinUCB`, IMLinUCB [49] and CUCB [47] for special graphs.

If we represent each edge by a $d$-dimensional feature vector, then we can generalize our `LT-LinUCB` for the large-scale case. The regret bound would become

$$O(\rho\gamma(G)D\sqrt{dmnT\log(1+Tn^2/d)}) = O(\gamma(G)Dd\sqrt{mnT}\ln(T))$$
$$= O(dn^3\sqrt{mT}\ln(T))$$

where $\rho = O(\sqrt{d\log(1+Tn^2/d)+2\log(1/\delta)})$. We have used $\|A_{t,u}\|^2 \le |N(u)|^2$ by assuming the feature vector all have L2-norm at most 1. The regret bound of IMLinUCB [49] under IC model with edge-level feedback is $O(C_G d\sqrt{mT}\ln(T)) = O(dmn\sqrt{T}\ln(T))$, which achieves $\sqrt{m}/n^2$ better order than ours.

**GOM property** Our GOM property (Theorem 1) plays a key role to bound the regret, similar to the TPM condition [47] in the IC model with edge-level feedback. Their proofs [47, 49] can be simplified by coupling the influence spread under weight $w$ and $w'$ to reduce the proof length significantly (see Appendix E). Under their setting, it is sufficient to prove the key property for monotone case $w \le w'$ since the confidence is estimated for each edge (base arm). The coupling technique can be designed so that the realized graph of $w$ is always a subgraph of $w'$. Then by comparing the connectivity difference in a subgraph, it is easy to derive the desired result.

Situations are different in our setting of node-level feedback. Since only group effect can be observed, we can not guarantee that the representative weight $w'$ is always larger than $w$ (see Section B for more discussions). Even though we can prove similar property for monotone $w \le w'$ and hope to generalize it to arbitrary $w, w'$ by leveraging $w \wedge w', w \vee w'$, it does not work. By leveraging $w \wedge w', w \vee w'$, the absolute function would be added to the edge-level (compared with the result formula of Theorem 1), while we can not observe single edges in group effect. Only the absolute functions on the differences of the weight sum are suitable for node-level feedback.

## B The Optimization Problem of Weight-Constrained IM

Recall that we have a confidence ellipsoid $\mathcal{C} = \{\mathcal{C}_v\}_{v\in V}$ with $\mathcal{C}_v = \{w'_v \in [0,1]^{|N(v)|} : \|w'_v - \hat{w}_v\|_{M_v} \le \rho_v\}$ and want to consider the optimization problem of weight-constrained influence maximization (WCIM):

$$\operatorname{argmax}_{(S,w'):S\in\mathcal{A},w'\in\mathcal{C}} r(S,w'). \tag{23}$$

Let $(S_{\mathcal{C}}^{\text{POpt}}, w_{\mathcal{C}}^{\text{POpt}})$ be the best solution. We want to find an $(\alpha, \beta)$-approximation oracle that outputs $(S', w')$ with $\mathbb{P}\left( r(S', w') \geq \alpha \cdot r(S_{\mathcal{C}}^{\text{POpt}}, w_{\mathcal{C}}^{\text{POpt}}) \right) \geq \beta$ for some $\alpha, \beta > 0$.

In the following, we first discuss the general difficulty, then give a general solution and later provide efficient methods for some special graph classes.

## B.1  General Difficulties

**The UCB-type method does not directly apply here**  Under the edge-level feedback of the IC model, the learner can update the information of each single edge if it is observed; then the confidence set of the unknown weight vector is just the direct product of the confidence interval over the edges:

$$\mathcal{C} = \mathcal{C}_1 \times \cdots \times \mathcal{C}_e \times \cdots \times \mathcal{C}_m \,,$$

where $\mathcal{C}_e = [L(e), U(e)]$ is 1-dimensional confidence interval of weight $w(e)$. Thus if we take the upper bound $U(e)$ of $\mathcal{C}_e$ for each edge $e$, the resulting vector $U = (U(e))_{e \in E}$ still lie in the confidence set $\mathcal{C}$ and any weight vector $w' \in \mathcal{C}$ satisfies $w' \leq U$. Since the reward function $r(S, w)$ is monotone increasing in weight vector $w$ (Lemma 9), the influence spread of any seed set $S$ under $U$ will be larger than $w'$. Hence $U$ would be the optimal weight vector for the WCIM problem (23). Then if we take the output $S_U$ from an usual $(\alpha, \beta)$-approximation `Oracle` for the IM with weight vector $U$, the pair $(S_U, U)$ is an $(\alpha, \beta)$-approximation solution for the problem WCIM. In such derivations, we have described a design of an $(\alpha, \beta)$-approximation `PairOracle`. This also explains why the designs in [49, 47] work.

Figure 2: An example that upper bound vector fails to lie in the confidence set.

But things are different in the node-level feedback of the LT model. In the node-level feedback, the learner can only observe group effects of edges instead of single edges, so the confidence set is high-dimensional ellipsoid instead of nice cuboid. If we take the upper bounds of each edge, which is equivalent to find the upper confidence bound of the vector $\chi(e)$ with respect to the confidence set $\mathcal{C}$, the resulting vector might jump out of the confidence set $\mathcal{C}$. Specifically, since `LT-LinUCB` updates the information of each single node if it has active in-neighbors, the confidence set $\mathcal{C}$ of the unknown weight vector $w$ is actually the direct product of the confidence set over the nodes:

$$\mathcal{C} = \mathcal{C}_1 \times \cdots \times \mathcal{C}_v \times \cdots \times \mathcal{C}_n \,,$$

where $\mathcal{C}_v$ is a $N(v)$-dimensional confidence set and is related to the edges with ending node $v$. Note that the confidence set $\mathcal{C}_v$ is different from the above $\mathcal{C}_e$ and we reuse the notation. For an example of 2-dimensional case (see Figure 2), there are two in-neighbors $u_1, u_2$ of $v$ and suppose a confidence ellipse has such a shape (the red ellipse). The vector of largest $w(e'_{u_1}, v), w(e'_{u_2}, v)$ (the red point) is not in the confidence set, and actually is far away from the confidence set. When more observations are collected, the red ellipse may shrink to the blue ellipse, but the vector of largest $w(e'_{u_1}, v), w(e'_{u_2}, v)$ (the blue point) just moves a little and its relative distance to the confidence set is even farther.

**Mixed integer optimization problem in bipartite graphs**  Consider the special bipartite graphs. The node set $V$ can be divided into $V_1$ and $V_2$ and each edge is from $V_1$ to $V_2$. Without loss of

generality, assume $|V_1| \geq K$, then a good solution $S$ must satisfy $S \subset V_1$. So the WCIM problem can be reformulated as

$$
\begin{aligned}
\max_{\alpha, w} \quad & \sum_{u \in V_1, v \in N^{\text{out}}(u)} \alpha(u)\, w(u, v) \\
s.t. \quad & \alpha(u) \in \{0, 1\} \text{ for any } u \in V_1 \\
& \sum_{u \in V_1} \alpha(u) \leq K \\
& w(u, v) \in [0, 1] \text{ for any } u \in V_1 \text{ and } v \in N^{\text{out}}(u) \\
& w(\cdot, v)^\top M_v\, w(\cdot, v) \leq \rho_v^2 \text{ for any } v \in V_2
\end{aligned}
\tag{24}
$$

where $w(\cdot, v) \in [0, 1]^{|N(v)|}$, $M_v \in \mathbb{R}^{|N(v)| \times |N(v)|}$ is some positive-definite matrix and $\rho_v$ is some constant.

This is a mixed integer optimization problem. Even if we relax the constraint of $\alpha(u) \in \{0, 1\}$ to $\alpha(u) \in [0, 1]$ to make the constraints convex, the objective is bilinear but not convex (or concave), making the problem hard to solve. This mixed integer programming is known to be difficult in the optimization field [41]. Some techniques of semidefinite programming (SDP) relaxations might be useful. We conjecture the approximation ratio, if solvable, is not constant and is $O(1/\ln(n))$ since there are roughly $n$ constraints for $w$, as also motivated by the greedy method for the problem of max vertex cover. We leave this as interesting future work.

If we write the vector $\alpha$ in a nice vector form, we can see the problem is a special maximum inner product [40, 37, 39]. This is an interesting direction but there are still many cases unexplored.

### B.2 $\epsilon$-net Method

The usual oracle for IM problem is to compute the seed set for a given weight vector. Now the confidence set $\mathcal{C}$ is a continuous set. A method is to discretize it. We can first find an $\epsilon$-net cover, compute the seed set by any usual oracle for each representative, and select the best pair. The complete method is provided in Algorithm 3. Recall that an $\epsilon$-net $\mathcal{N}$ for a set $\mathcal{C}$ is for any $w' \in \mathcal{C}$, there exists a $\pi(w') \in \mathcal{N}$ such that $\|w' - \pi(w')\|_2 \leq \epsilon$. The minimal size of possible $\mathcal{N}$ is denoted as $N_{\mathcal{C}, \epsilon}$.

---

**Algorithm 3** $\epsilon$-net `PairOracle`

1: **Input:** Confidence ellipsoid $\mathcal{C}$; offline IM `Oracle`; seed set cardinality $K$; parameter $\epsilon$
2: Find an optimal $\epsilon$-net $\mathcal{N}$ for $\mathcal{C}$ with size $N_{\mathcal{C}, \epsilon}$
3: **for** $\pi \in \mathcal{N}$ **do**
4:     Compute the seed set $S_\pi$ and $r(S_\pi, \pi)$ by `Oracle` with $\pi$ and $K$
5: **end for**
6: **Output:** $(S', w') = \operatorname{argmax}_{(S_\pi, \pi) : \pi \in \mathcal{N}} r(S_\pi, \pi)$

---

Then we have the following approximation guarantee for the $\epsilon$-net method.

**Lemma 3.** *The Algorithm 3 runs with confidence ellipsoid $\mathcal{C}$, seed set cardinality $K$, parameter $\epsilon$ and an $(\alpha', \beta')$-approximation* `Oracle`. *Then its output satisfies*

$$
\mathbb{P}\left( r(S', w') \geq \alpha \cdot r(S_{\mathcal{C}}^{\text{POpt}}, w_{\mathcal{C}}^{\text{POpt}}) \right) \geq \beta,
$$

*where $\alpha = \alpha'\left(1 - \frac{mn \cdot \epsilon}{K}\right)$ and $\beta = \beta'$. Thus the $\epsilon$-net* `PairOracle` *(Algorithm 3) is $(\alpha, \beta)$-approximation.*

*Proof.* For any $w' \in \mathcal{C}$, let $\pi(w') \in \mathcal{N}$ be its representative such that $\|w' - \pi(w')\|_2 \leq \epsilon$. Let $S_{w'}^*$ denote the output of `Oracle` with input $w'$, then $\mathbb{P}\left( r(S_{w'}^*, w') \geq \alpha' \cdot \text{Opt}_{w'} \right) \geq \beta'$. Thus

$$
\begin{aligned}
r\left( S_{\mathcal{C}}^{\text{POpt}}, w_{\mathcal{C}}^{\text{POpt}} \right) &\leq r\left( S_{\mathcal{C}}^{\text{POpt}}, \pi\left( w_{\mathcal{C}}^{\text{POpt}} \right) \right) + mn \cdot \epsilon \\
&\leq \text{Opt}_{\pi\left( w_{\mathcal{C}}^{\text{POpt}} \right)} + mn \cdot \epsilon
\end{aligned}
$$

$$\leq \frac{1}{\alpha'} r\left(S^*_{\pi(w_\mathcal{C}^{\mathrm{POpt}})}, \pi\left(w_\mathcal{C}^{\mathrm{POpt}}\right)\right) + mn \cdot \epsilon$$

$$\leq \frac{1}{\alpha'} r(S', w') + mn \cdot \epsilon \,,$$

where the first inequality is by Lipschitz continuity of $r$ (Lemma 10) and $\left\| w_\mathcal{C}^{\mathrm{POpt}} - \pi(w_\mathcal{C}^{\mathrm{POpt}}) \right\|_2 \leq \epsilon$, the third inequality is by the definition of `Oracle` and holds with probability at least $\beta'$ and the last inequality is by the rule of Algorithm 3.

Hence with probability at least $\beta'$,

$$r(S', w') \geq \alpha' \cdot \left( r(S_\mathcal{C}^{\mathrm{POpt}}, w_\mathcal{C}^{\mathrm{POpt}}) - mn \cdot \epsilon \right) \geq \alpha' \left( 1 - \frac{mn \cdot \epsilon}{K} \right) \cdot r(S_\mathcal{C}^{\mathrm{POpt}}, w_\mathcal{C}^{\mathrm{POpt}}) \,,$$

where the second inequality is by $r(S_\mathcal{C}^{\mathrm{POpt}}, w_\mathcal{C}^{\mathrm{POpt}}) \geq K$. $\qquad\square$

The minimal size $N_{\mathcal{C},\epsilon}$ of the $\epsilon$-net for the $m$-dimensional ellipsoid $\mathcal{C}$ has order $\Theta((1/\epsilon)^m)$, which is exponential in $\epsilon$. So this method, though accurate, is not very efficient.

### B.3 Graphs with In-degree at Most 1

We discuss the method to solve the case of graphs that any node has at most one incoming edge. This includes examples in Figure 3. For such graphs, the node-level feedback is actually edge-level feedback. More specifically, our `LT-LinUCB` will update the information of each single edge if its start node is active. Thus the confidence set $\mathcal{C}$ is the direct product of the confidence intervals of each edge, similar to IC model with edge-level feedback.

Figure 3: Examples of graphs with in-degree at most 1. (a) bar graph. (b) chain graph. (c) out-arborescence graph. (d) out-star graph. (e) certain bipartite graph. Each undirected edge represents a pair of directed edges pointing to opposite directions.

Hence we can just perform as [11, 49]. As mentioned above, we first take the upper bound for each edge and formulate $U$, then use an $(\alpha, \beta)$-approximation `Oracle` to compute $S_U$ such that

$$\mathbb{P}\left( r(S_U, U) \geq \alpha \cdot r(S_\mathcal{C}^{\mathrm{POpt}}, w_\mathcal{C}^{\mathrm{POpt}}) \right) \geq \beta \,. \tag{25}$$

So we get an efficient $(\alpha, \beta)$-approximation `PairOracle` for these special graphs.

### B.4 Bipartite Graphs

We consider the special case of bipartite graphs here where there are two node sets $V_1$ and $V_2$ and each edge is from $V_1$ to $V_2$ (see Figure 4 for examples). This is a popular influence spread formulation for one step and is a generalization of vertex cover.

Recall that the objective is to solve

$$\max_{S \in \mathcal{A}, w' \in \mathcal{C}} r(S, w') = \max_{S \in \mathcal{A}} \max_{w' \in \mathcal{C}} r(S, w') \,.$$

Note that $r(S, w) = \sum_{u \in S, v \in N^{\mathrm{out}}(u)} w(e_{u,v})$ is linear in $w$ for the bipartite graphs. Let

$$r(S) := \max_{w' \in \mathcal{C}} r(S, w') = \max_{w' \in \mathcal{C}} \sum_{u \in S, v \in N^{\mathrm{out}}(u)} w(e_{u,v}) \,.$$

Figure 4: Examples of special bipartite graphs. (a) bipartite graph with in-degree at most 2. (b) bipartite graph with in-degree 3.

Recall that $\mathcal{C} = \{w' \in \mathbb{R}^m : \|w' - \hat{w}\|_M \leq \rho\}$ for some positive-definite matrix $M$ and a constant $\rho \geq 0$. So the computation of $r(S)$ is quadratic constrained linear programming and can be solved efficiently. We have the following properties for $r(S)$. The first one is about its monotonicity for any graph.

**Lemma 4.** *For any graph, given the confidence set $\mathcal{C}$, the function $r(S) = \max_{w' \in \mathcal{C}} r(S, w')$ is monotone increasing in $S$. That is, $r(S) \leq r(S')$ if $S \subseteq S'$.*

*Proof.* Let $r(S) = r(S, w_S), r(S') = r(S', w_{S'})$. Then

$$r(S) = r(S, w_S) \leq r(S', w_S) \leq r(S', w_{S'}) = r(S') \,.$$

$\square$

The next one states the submodularity of $r(S)$ for bipartite graphs with in-degree at most 2 (for example Figure 4(a)).

**Lemma 5.** *In bipartite graphs with in-degree at most 2, the function $r(S) = \max_{w' \in \mathcal{C}} r(S, w')$ satisfies submodularity. That is, for arbitrary set $S \subseteq S'$ and node $u \notin S'$, there is*

$$r(S \cup \{u\}) - r(S) \geq r(S' \cup \{u\}) - r(S) \,. \tag{26}$$

*Proof.* As discussed in Section B.1, the confidence set $\mathcal{C}$ for unknown weight vector $w$ is actually the direct product of confidence set over nodes, that is

$$\mathcal{C} = \Pi_{v \in V_2} \, \mathcal{C}_v \,,$$

where $\mathcal{C}_v$ is the confidence set for in-coming edges of node $v$. The edges in each $\mathcal{C}_v$ are disjoint with each other, so

$$r(S) = \max_{w' \in \mathcal{C}} \sum_{u \in S, v \in N^{\text{out}}(u)} w'(e_{u,v}) = \max_{w' \in \mathcal{C}} \sum_{v \in V_2} \sum_{u \in S, u \in N^{\text{in}}(v)} w'(e_{u,v})$$

$$= \sum_{v \in V_2} \max_{w'_v \in \mathcal{C}_v} \sum_{u \in S, u \in N^{\text{in}}(v)} w'_v(e_{u,v}) \,,$$

where $w'_v = w'(e_{u,v})_{u \in N^{\text{in}}(v)}$. So to maximize over $\mathcal{C}$, it suffices to maximize the weights of incoming edges for each $v \in V_2$.

Since each node has at most two in-coming edges, if $e_{u,v} \in E$ for some $v$ then it must hold that there is at most one in-neighbor of $v$ from $S'$. For node $v$ such that $e_{u,v} \in E$ but there is no edge from $S'$ to $v$, the contribution of $v$'s part to $S, S'$ are the same.

For node $v$ such that $e_{u,v}, e_{u',v} \in E$ for some $u' \in S' \setminus S$, it suffices to prove that

$$\max_{w'_v \in \mathcal{C}_v} w'_v(e_{u,v}) \geq \max_{w'_v \in \mathcal{C}_v} \{w'_v(e_{u,v}) + w'_v(e_{u',v})\} - \max_{w'_v \in \mathcal{C}_v} w'_v(e_{u',v}) \,,$$

which is obviously true.

For node $v$ such that $e_{u,v}, e_{u',v} \in E$ for some $u' \in S \subseteq S'$, the contribution of $v$'s part to $S, S'$ are the same. $\square$

---
**Algorithm 4** Greedy `PairOracle`
---
1: **Input:** Graph $G = (V, E)$, confidence set $\mathcal{C}$, seed set cardinality $K$
2: **Initialize:** $S = \emptyset$
3: **for** $i \in [K]$ **do**
4:     $v = \text{argmax}_{u \in V \setminus S} \, r(S \cup \{u\}) - r(S)$
5:     $S = S \cup \{v\}$
6: **end for**
7: Output $S$

---

With the submodularity property, we can get the approximation result by designing a greedy policy (Algorithm 4).

**Lemma 6.** *Recall that $S_{\mathcal{C}}^{\text{POpt}} = \text{argmax}_{S \in \mathcal{A}} \max_{w' \in \mathcal{C}} r(S, w') = \text{argmax}_{S \in \mathcal{A}} r(S)$ is the optimal seed set given confidence set $\mathcal{C}$. Let $S'$ be the solution returned by Greedy `PairOracle` (Algorithm 4). Then for bipartite graphs with in-degree at most 2,*

$$r(S') \geq \left(1 - \frac{1}{e}\right) r(S_{\mathcal{C}}^{\text{POpt}}). \tag{27}$$

The proof is a direct application of [19, Theorem 2.1] by noting that the function $r(\cdot)$ satisfies monotonicity (Lemma 4) and submodularity (Lemma 5) in such graphs.

**A counterexample of in-degree** 3    Here we show an example of bipartite graphs with in-degree 3 but the $r(\cdot)$ does not have the submodularity property.

Let $V_1 = \{u_1, u_2, u_3\}$, $|V_2| = 1$ and there are only 3 edges (see Figure 4(b) for example). The confidence set $\mathcal{C} = \left\{ w' \in \mathbb{R}^3 : \|w'\|_M \leq 1 \right\}$ with

$$M = \begin{bmatrix} 2 & 1 & 0 \\ 1 & 3 & 1 \\ 0 & 1 & 2 \end{bmatrix}.$$

Note $M = I + (1, 1, 0)^\top (1, 1, 0) + (0, 1, 1)^\top (0, 1, 1)$ can happen for our algorithm `LT-LinUCB`. We solve the optimization problem and get $r(\{u_1, u_2\}) \approx 0.791, r(\{u_2, u_3\}) \approx 0.791, r(\{u_2\}) \approx 0.707, r(\{u_1, u_2, u_3\}) \approx 1.000$. Thus let $u = u_1, S = \{u_2\}, S' = \{u_2, u_3\}$, we have

$$r(S \cup \{u\}) - r(S) < 0.09 < 0.2 < r(S' \cup \{u\}) - r(S'),$$

which violates the definition of submodularity.

### B.5   Directed Acyclic Graphs

Recall that $r(S) = \max_{w' \in \mathcal{C}} r(S, w')$ and $S_{\mathcal{C}}^{\text{POpt}} = \text{argmax}_{S \in \mathcal{A}} r(S)$ is the optimal solution. Let $S'$ be the output of Greedy `PairOracle` (Algorithm 4). Then we have the following $1/K$-approximation result.

**Lemma 7.** *For general graphs, suppose we can compute $r(S)$ for any $S$. Then*

$$r(S') \geq \frac{1}{K} \cdot r(S_{\mathcal{C}}^{\text{POpt}}). \tag{28}$$

*Proof.* Denote $S_{\mathcal{C}}^{\text{POpt}} = \{s_1^*, s_2^*, ..., s_K^*\}$. Assume the Greedy `PairOracle` first chooses $s'$. Then $s' \in S'$ and $s' = \text{argmax}_{u \in V} r(\{u\})$ or equivalently $r(\{s'\}) \geq r(\{u\})$ for any $u \in V$. By monotonicity of $r$ (Lemma 4),

$$r(S') \geq r(\{s'\}) \geq \frac{1}{K} \cdot (r(\{s_1^*\}) + r(\{s_2^*\}) + ... + r(\{s_K^*\})) .$$

It suffices to prove that $r$ satisfies the subadditivity. It is well known that the reward function $r(\cdot, w')$ satisfies submodularity in LT model [19]. Then for any $S \subseteq S''$,

$$r(S, w') + r(S'', w') \geq r(S \cup S'', w') + r(S \cap S'', w') \geq r(S \cup S'', w') .$$

Recall that $r(S_{\mathcal{C}}^{\mathrm{POpt}}) = r(S_{\mathcal{C}}^{\mathrm{POpt}}, w_{\mathcal{C}}^{\mathrm{POpt}})$. Then

$$
\begin{aligned}
r(\{s_1^*\}) &+ r(\{s_2^*\}) + \ldots + r(\{s_K^*\}) \\
&\geq r(\{s_1^*\}, w_{\mathcal{C}}^{\mathrm{POpt}}) + r(\{s_2^*\}, w_{\mathcal{C}}^{\mathrm{POpt}}) + \ldots + r(\{s_K^*\}, w_{\mathcal{C}}^{\mathrm{POpt}}) \\
&\geq r(S_{\mathcal{C}}^{\mathrm{POpt}}, w_{\mathcal{C}}^{\mathrm{POpt}}) = r(S_{\mathcal{C}}^{\mathrm{POpt}})
\end{aligned}
$$

and the result follows. $\qquad\square$

Next we show that for directed acyclic graphs (DAGs), there is an efficient method to compute $r(S) = \max_{w' \in \mathcal{C}} r(S, w')$.

---

**Algorithm 5** Compute $r(S)$ in DAGs

---

1: **Input:** DAG $G = (V, E)$; seed set $S$;
   the set of confidence ellipsoids $(\mathcal{C}_v)_{v \in V}$ with $\mathcal{C}_v = \{w'_v : (w'_v)^\top M_v w'_v \leq \rho_v^2\}$
2: **Initialize:** Delete all in-edges to nodes in $S \subseteq V$
3: Use topological ranking to form 'layers' of nodes $L_0, \ldots, L_\ell, \ldots, L_{n-1}$ satisfies any edge $e \in E$ points from $L_i$ to $L_j$ for some $i < j$
4: $r_S^u = 1$ for $u \in S$; $r_S^u = 0$ for $u \in L_0 \setminus S$
5: **for** $\ell = 1, 2, \ldots$ **do**
6:     **for** $v \in L_\ell$ **do**
7:         Solve $r_S^v = \max_{w' \in \mathcal{C}_v} \sum_{u \in N(v)} r_S^u \cdot w'(e_{u,v})$
8:     **end for**
9: **end for**
10: **Output:** $r(S) = \sum_{v \in V} r_S^v$

---

For seed set $S$, delete all in-coming edges to $S$. Take all nodes with in-degree 0 and form a set $L_0 \supseteq S$. Then consider the reduced subgraph for remaining nodes $V \setminus L_0$, take all nodes in the subgraph with in-degree 0 and form a set $L_1$. Note subgraphs of DAGs are still DAGs and in DAGs there are nodes with in-degree 0, otherwise we could find a cycle by adaptively adding in-neighbors. Then the procedure can continue until no node is left. Such process is just topological ranking to form 'layers' of nodes. For any node $u \in L_\ell$, its incoming edges are all from previous layers (except seed nodes), or equivalently nodes in $L_0 \cup L_1 \cup \cdots \cup L_{\ell-1}$. There are at most $n$ layers.

Let $E'_\ell$ to denote the edges that has end node in layer $\ell$ and $E'_{\ell:\ell'} = E'_\ell \cup E'_{\ell+1} \cup \cdots \cup E'_{\ell'}$. Then $E'_\ell \bigcap E'_{\ell'} = \emptyset$ if $\ell \neq \ell'$.

Let $r_S^u(w')$ be the probability that node $u$ will be influenced under the weight vector $w'$ when the seed set is $S$ and $r_S^u = \max_{w' \in \mathcal{C}} r_S^u(w')$. For seed node $u \in S$, it is activated with probability 1, or $r_S^u(w') \equiv 1$. For node $u \in L_0 \setminus S$, there is no directed path connecting from seed node $S$, so its activation probability is always 0, or $r_S^u(w') \equiv 0$. So we have computed $r_S^u$ for $u \in L_0$.

Let $\ell = 1$. For node $u \in L_\ell$, its incoming edges all come from former layers $< \ell$. Note that $r_S^u$ has been defined for any layer $< \ell$ and $r(S, w')$ can be decomposed as

$$
\sum_{v \in L_\ell} \left( \sum_{u \in N^{\mathrm{in}}(v)} r_S^u \cdot w'(e_{u,v}) \right) \cdot f_v(w', E'_{\ell+1:n}), \tag{29}
$$

where $f_v(w', E'_{\ell+1:n})$ is the expected influenced nodes by node $v$ for *later layers* and it only relates with the edges ending in later layers. Recall that the constraints are added to the edges with the same ending node. The edge $e_{u,v}$ for $v \in L_\ell$ ends in $L_\ell$, so it is independent with $E'_{\ell'}$ for $\ell' > \ell$. Note that $f_v(w', E'_{\ell+1:n}) \geq 1 > 0$ since node $v$ at least influences itself. So to maximize $r(S, w')$ over $w' \in \mathcal{C}$, the weights related with edges in $E'_\ell$ can be maximized separately. Specifically, we can solve the maximization problem for each $v \in L_\ell$:

$$
\max \sum_{u \in N^{\mathrm{in}}(v)} r_S^u \cdot w'(e_{u,v}) \tag{30}
$$

$$
s.t. \quad (w'_v)^\top M_v w'_v \leq \rho_v^2
$$

where the $w'_v = (w'(e_{u,v}))_{u \in N(v)}$, $M_v$ is some positive-definite matrix and $\rho_v$ is some constant. This optimization problem is linear programming with quadratic convex constraints and can be solved efficiently. The resulting maximum value is actually $r_S^v$. So we can compute $r_S$ for layer $\ell$. Then we can compute $r(S) = \sum_{v \in V} r_S^v$ by repeating steps (29) (30) with induction on $\ell$. The process is presented in Algorithm 5.

The key point to make this through for DAGs is based on the linearity of LT. Then we can decompose the objective functions to *isolated* parts and use common optimization methods to solve each part step by step.

## C  Analysis of `OIM-ETC` Algorithm

We first provide the regret bound of `OIM-ETC` Algorithm under both IC and LT models and then give discussions about it.

### C.1  Proof of Theorem 3

Recall that $\hat{w}$ is the empirical estimate of weight vector $w$ (line 6 of Algorithm 2) and $\hat{S}$ is the output of the $(\alpha, \beta)$-approximation `Oracle` under estimated weight vector $\hat{w}$ (line 7 of Algorithm 2). Define event

$$\mathcal{F} = \left\{ r(\hat{S}, \hat{w}) < \alpha \cdot \mathrm{Opt}_{\hat{w}} \right\} .$$

Then $\mathbb{P}(\mathcal{F}) < 1 - \beta$ since the `Oracle` is $(\alpha, \beta)$-approximation.

We first decompose the regret

$$
\begin{aligned}
R(T) &= \mathbb{E}\left[ \sum_{t=1}^{T} (\alpha\beta \cdot \mathrm{Opt}_w - r(S_t, w)) \right] \\
&= \mathbb{E}\left[ \sum_{t=1}^{nk} (\alpha\beta \cdot \mathrm{Opt}_w - r(S_t, w)) \right] + \mathbb{E}\left[ \sum_{t=nk+1}^{T} (\alpha\beta \cdot \mathrm{Opt}_w - r(S_t, w)) \right] \\
&\le nk\Delta_{\max} + (T - nk)\mathbb{E}\left[ \alpha\beta \cdot \mathrm{Opt}_w - r(\hat{S}, w) \right] \\
&\le nk\Delta_{\max} + (T - nk)\beta \cdot \mathbb{E}\left[ \alpha \cdot \mathrm{Opt}_w - r(\hat{S}, w) \Big| \mathcal{F}^c \right]
\end{aligned}
\tag{31}
$$

where the last inequality is by

$$\mathbb{E}\left[ r(\hat{S}, w) \right] = \mathbb{E}\left[ r(\hat{S}, w) \Big| \mathcal{F} \right] \mathbb{P}(\mathcal{F}) + \mathbb{E}\left[ r(\hat{S}, w) \Big| \mathcal{F}^c \right] \mathbb{P}(\mathcal{F}^c) \ge \beta \cdot \mathbb{E}\left[ r(\hat{S}, w) \Big| \mathcal{F}^c \right] .$$

Note under $\mathcal{F}^c$,

$$
\begin{aligned}
\alpha \cdot \mathrm{Opt}_w &= \alpha \cdot r(S_w^{\mathrm{Opt}}, w) \\
&\le \alpha \cdot r(S_w^{\mathrm{Opt}}, \hat{w}) + \alpha \cdot mn \cdot \max_{e \in E} |\hat{w}(e) - w(e)| \\
&\le \alpha \cdot r(S_{\hat{w}}^{\mathrm{Opt}}, \hat{w}) + \alpha \cdot mn \cdot \max_{e \in E} |\hat{w}(e) - w(e)| \\
&\le r(\hat{S}, \hat{w}) + \alpha \cdot mn \cdot \max_{e \in E} |\hat{w}(e) - w(e)| \\
&\le r(\hat{S}, w) + (1 + \alpha) \cdot mn \cdot \max_{e \in E} |\hat{w}(e) - w(e)| .
\end{aligned}
\tag{32}
$$

Then when $\max_{e \in E} |\hat{w}(e) - w(e)| < \frac{\Delta_{\min}}{(1+\alpha)mn} =: \epsilon_0$, $\hat{S} \notin \mathcal{S}_B$. So the regret becomes

$$
\begin{aligned}
R(T) &\le nk\Delta_{\max} + (T - nk) \cdot 2m \exp(-2k\epsilon_0^2)\Delta_{\max} \\
&\le \left( nk + 2mT \exp(-2k\epsilon_0^2) \right) \Delta_{\max} \\
&= \frac{n\Delta_{\max}}{2\epsilon_0^2} \ln^+ \frac{4mT\epsilon_0^2}{n} + \frac{n\Delta_{\max}}{2\epsilon_0^2}
\end{aligned}
$$

where the first inequality is to bound the complement of the event $\max_{e \in E} |\hat{w}(e) - w(e)| < \epsilon_0$ by the Chernorff-Hoeffding bound (Lemma 8), the equality is optimized with $k$ satisfying $\exp(2k\epsilon_0^2) = 4mT\epsilon_0^2/n$ and $\ln^+(x) = \max\{0, \ln(x)\}$.

Therefore taking $k = \max\left\{1, \frac{1}{2\epsilon_0^2} \ln \frac{4mT\epsilon_0^2}{n}\right\} = \max\left\{1, \frac{2m^2n^2}{\Delta_{\min}^2} \ln \frac{T\Delta_{\min}^2}{mn^3}\right\}$ together with $R(T) \leq T\Delta_{\max}$, the regret satisfies

$$R(T) \leq \min\left\{T\Delta_{\max}, n\Delta_{\max} + \frac{2m^2n^3\Delta_{\max}}{\Delta_{\min}^2}\left(1 + \ln^+ \frac{T\Delta_{\min}^2}{mn^3}\right)\right\}$$

$$= O\left(\frac{m^2n^3\Delta_{\max}}{\Delta_{\min}^2} \ln(T)\right). \tag{33}$$

Next we prove the problem-independent bound. Following (32) under $\mathcal{F}^c$, with a suitable $\epsilon$ to be decided later,

$$\mathbb{E}\left[\alpha \cdot \mathrm{Opt}_w - r(\hat{S}, w)\right] \leq 2mn \cdot \mathbb{E}\left[\max_{e \in E} |\hat{w}(e) - w(e)|\right]$$

$$\leq 2mn\epsilon + 2mn \sum_{s=0}^{\infty} 2^{s+1}\epsilon \cdot \mathbb{P}\left(2^s\epsilon < \max_{e \in E} |\hat{w}(e) - w(e)| \leq 2^{s+1}\epsilon\right)$$

$$\leq 2mn\epsilon + 2mn \sum_{s=0}^{\infty} 2^{s+1}\epsilon \cdot \mathbb{P}\left(\exists e \in E, |\hat{w}(e) - w(e)| > 2^s\epsilon\right)$$

$$\leq 2mn\epsilon + 2mn \sum_{s=0}^{\infty} 2^{s+1}\epsilon \cdot 2m\exp(-2k2^{2s}\epsilon^2)$$

$$= 2mn\epsilon + \frac{8m^2n}{\sqrt{2k}} \sum_{s=0}^{\infty} \sqrt{2k}2^s\epsilon \cdot \exp(-(\sqrt{2k}2^s\epsilon)^2).$$

Let $X_s := \sqrt{2k}2^s\epsilon$. Note that the function $f(x) = xe^{-x^2}$ increases in $[0, 1/\sqrt{2}]$ and decreases in $[1/\sqrt{2}, \infty)$. Let $s_0$ satisfy

$$2^{s_0} < \frac{1}{2\sqrt{k}\epsilon} \leq 2^{s_0+1},$$

or equivalently $X_s < 1/\sqrt{2}$ for $s \leq s_0$ and $X_s \geq 1/\sqrt{2}$ for $s \geq s_0 + 1$. Then we can divide the sum into three parts

$$\sum_{s=0}^{s_0-1} f(X_s) + \sum_{s=s_0}^{s_0+1} f(X_s) + \sum_{s_0+2}^{\infty} f(X_s).$$

By monotonicity, $\sum_{s=0}^{s_0-1} f(X_s) \leq \int_0^{s_0} f(x)\,dx$ and $\sum_{s_0+2}^{\infty} f(X_s) \leq \int_{s_0+1}^{\infty} f(x)\,dx$. Thus

$$\mathbb{E}\left[\alpha \cdot \mathrm{Opt}_w - r(\hat{S}, w)\right] \leq 2mn\epsilon + \frac{8m^2n}{\sqrt{2k}}\left(\int_0^{\infty} f(x)\,dx + f(X_{s_0}) + f(X_{s_0+1})\right)$$

$$\leq 2mn\epsilon + \frac{8m^2n}{\sqrt{2k}}\left(\frac{1}{2} + 2f(1/\sqrt{2})\right)$$

$$= 2mn\epsilon + \frac{8m^2n}{\sqrt{2k}}\left(\frac{1}{2} + \sqrt{2}\exp(-1/2)\right)$$

$$\leq 2mn\epsilon + 7.69m^2n/\sqrt{k}.$$

By substituting it to (31), the regret is bounded by

$$R(T) \leq nk\Delta_{\max} + T \cdot \left(2mn\epsilon + 7.69m^2n/\sqrt{k}\right)$$

$$\leq n^2k + T \cdot \left(2mn\epsilon + 7.69m^2n/\sqrt{k}\right)$$

$$\leq 3.9(mn)^{4/3}T^{2/3} + 1 \leq 5(mn)^{4/3}T^{2/3}$$
$$= O\left((mn)^{4/3}T^{2/3}\right), \tag{34}$$

where we take $k = 3.9m^{4/3}n^{-2/3}T^{2/3}$ and $\epsilon = 1/(2mnT)$.

## C.2 Discussions

As we mentioned, our `OIM-ETC` algorithm is model independent and applies to both LT and IC model with node-level feedback.

Recall that for a typical influence spread under the IC model, each edge $e$ is *live* with the associated probability $w(e) \in [0, 1]$ and a node is activated if there is a (directed) path connecting from the seed set. For the IC model, there are three types of feedback: (1) bandit feedback, where the learner can only observe the number of influenced nodes; (2) edge-level feedback, where the learner can observe the liveness status of each outgoing edge from the activated nodes; (3) node-level feedback, where the learner can only observe the spread propagation but not individual edge liveness. The bandit feedback presents the least information and is most difficult considering the nonlinearity and complexity of the influence reward function. The edge-level feedback gives the most informative feedback and most previous work study this scheme [11, 47, 49, 50].

Since our `OIM-ETC` only selects size-1 seed set in the exploration phase, so the node-level feedback of the first-step triggering is the same with the edge-level feedback. Thus `OIM-ETC` can be applied to both IC and LT model. Though simple, `OIM-ETC` is the first model-independent algorithm for OIM[3]. Furthermore, the computational complexity for `OIM-ETC` is really low, as it only calls once of the offline oracle.

As mentioned in related work, the algorithm for the combinatorial partial monitoring [32] can be applied in OIM for both LT and IC models with node-level feedback. However, the second best solution used in their algorithm could not be directly computed in the offline IM setting. Hence only their the second stop-exploration condition applies and a regret bound of $O(nm^{3/2}T^{2/3}\ln(T))$ is obtained. Our `OIM-ETC` is better in $O(\ln(T))$ term and a bit worse in $O((n/\sqrt{m})^{1/3})$ term. Also our `OIM-ETC` has a problem-dependent regret bound.

Comparing with `LT-LinUCB` we see that `OIM-ETC` only requires the first-step node feedback, not the full diffusion process feedback of $S_{t,0}, S_{t,1}, \ldots, S_{t,\tau}, \ldots$. Moreover, it only requires the offline oracle to solve the maximization problem using the empirical mean as the fixed weight vector. The objective function in this case is known to be monotone and submodular [19, 34], and thus a greedy algorithm [19] or IMM algorithm [42] could achieve $1 - 1/e - \varepsilon$ approximation (for any small $\varepsilon > 0$) with probability at least $1 - 1/n$. That is, $(\alpha, \beta)$-approximation `Oracle` with $\alpha = 1 - 1/e - \varepsilon$ and $\beta = 1 - 1/n$ has an efficient implementation. This is also easier than the `PairOracle`, which has the confidence ellipsoid as the constraint on weight vectors.

## D Technical Lemmas

**Lemma 8.** *(Chernoff-Hoeffding bound) Let $X_1, X_2, \ldots, X_n$ be independent random variables with common support $[0, 1]$. Let $S_n = X_1 + X_2 + \cdots + X_n$ and $\mu = \mathbb{E}[S_n]$. Then for any $\epsilon \geq 0$,*

$$\mathbb{P}(|S_n - \mu| \geq n\epsilon) \leq 2\exp\left(-2n\epsilon^2\right).$$

Next is a property of the reward function on the weight vector under the LT model. Note that the similar property also holds for IC model [11, Lemma 6].

**Lemma 9.** *Under the LT model, the reward function $r(S, w)$ is monotone increasing in $w$. And for any seed set $S$ and any two weight vectors $w, w' \in [0, 1]^m$, there is*

$$|r(S, w) - r(S, w')| \leq mn \cdot \max_{e \in E} |w(e) - w'(e)|. \tag{35}$$

*Proof.* We first prove the monotonicity. Suppose $w(e) \le w'(e)$ for all $e \in E$. For any fixed thresholds $\theta_v$'s, the instance of influence graph under weight vector $w$ is always a subgraph of $w'$ since any activated node $v$ under $w$ is always activated under $w'$. Thus $r(S, w) \le r(S, w')$.

For (35), it is enough to prove the case $w \le w'$; otherwise we can prove it first for $w \wedge w'$ and $w \vee w'$ and then conclude the result since $r(S, w \wedge w') \le \{r(S, w), r(S, w')\} \le r(S, w \vee w')$.

Now assume $w, w'$ only differ on one edge $e$: $w'(e) > w(e)$ and $w'(e') = w(e')$ for any $e' \ne e$. For any fixed thresholds $\theta_v$'s, consider the two diffusion process under $w, w'$. If the spreads are different, then the starting node that the diffusion processes starts to become different must be the end node of edge $e$. Then this difference would cause at most $n$ nodes differences. Such an event happens when the difference of $w'(e) - w(e)$ contributes to the activation of end node of edge $e$, which has probability at most $w'(e) - w(e)$. Thus $r(S, w') - r(S, w) \le n \cdot (w'(e) - w(e))$.

Then for vectors $w \le w'$, we can construct at most $|E| = m$ vector pairs from $w$ to $w'$ with each pair only differing on one edge. By summing them up, we get $r(S, w') - r(S, w) \le mn \cdot \max_{e \in E} |w(e) - w'(e)|$. $\qquad\square$

**Lemma 10.** *For any seed set $S$ and any two weight vectors $w, w' \in [0, 1]^m$, there is*

$$|r(S, w) - r(S, w')| \le mn \cdot \|w - w'\|_2 .$$

*Proof.* Lemma 10 can be concluded directly from Lemma 9 since it is obvious that $\max_{e \in E} |w(e) - w'(e)| \le \|w - w'\|_2$. $\qquad\square$

# E  A Simplified Proof for the TPM Condition

We give a simplified proof for the TPM condition under the IC model with edge-level feedback, which corresponds to [49, Theorem 3] and the key equation [47, Lemma 2, (28)]. For completeness, we also give the theorem statement here, which mainly follow the notations of [49].

$f(S, w, v)$ is the influence probability of seed set $S$ to node $v$ when the mean of the weights is vector $w$. $O(e)$ denotes the event that edge $e$ is observed. Recall that an edge $e$ is relevant with $S, v$ means there exists a path $\ell$ from a seed node $s \in S$ to $v$ such that (1) $e \in \ell$ and (2) $\ell$ does not contain another seed node other than $s$. In the following, we use boldface $\mathbf{w}$ to represent a random realization of the weight vector.

**Theorem 4.** *(restated) For any node $v \notin S$,*

$$f(S, U, v) - f(S, \bar{w}, v) \le \sum_{\substack{e \text{ is relevant with } S, v}} \mathbb{E}_{\bar{w}}[\mathbb{1}\{O(e)\} \cdot (U(e) - \bar{w}(e)) \mid S] \qquad (36)$$

*Proof.* Note that

$$f(S, U, v) = \mathbb{E}_{\mathbf{w}_1 \sim U} \mathbb{1}\{v \text{ is influenced under } \mathbf{w}_1\} ,$$
$$f(S, \bar{w}, v) = \mathbb{E}_{\mathbf{w}_2 \sim \bar{w}} \mathbb{1}\{v \text{ is influenced under } \mathbf{w}_2\} .$$

When we compute the difference of these two terms, we do not need to make these two $\mathbf{w}$ independent. Specifically, for each edge $e$, we can design $\mathbf{w}_1, \mathbf{w}_2$ in the following way. Suppose for each edge $e$ we independently draw a uniform random variable $A(e)$ over $[0, 1]$, let

$$
\begin{aligned}
\mathbf{w}_1(e) = \mathbf{w}_2(e) = 1, & \qquad\qquad \text{if } A(e) \le \bar{w}(e) \,; \\
\mathbf{w}_1(e) = 1, \mathbf{w}_2(e) = 0, & \qquad\qquad \text{if } A(e) \in (\bar{w}(e), U(e)] \,; \\
\mathbf{w}_1(e) = \mathbf{w}_2(e) = 0, & \qquad\qquad \text{if } A(e) > U(e) \,.
\end{aligned}
$$

Such a design of $\mathbf{w}_2$ would introduce a subgraph of $\mathbf{w}_1$ and the marginal expected means of $\mathbf{w}_1, \mathbf{w}_2$ are $U, \bar{w}$ respectively. Then the difference would become much simpler

$$f(S, U, v) - f(S, \bar{w}, v) = \mathbb{E}_{\mathbf{w}_1, \mathbf{w}_2 \sim A}[f(S, \mathbf{w}_1, v) - f(S, \mathbf{w}_2, v)]$$

and $f(S, \mathbf{w}_1, v) - f(S, \mathbf{w}_2, v) = 0$ or $1$.

$f(S, \mathbf{w}_1, v) - f(S, \mathbf{w}_2, v) = 1$ means $f(S, \mathbf{w}_1, v) = 1$ and $f(S, \mathbf{w}_2, v) = 0$. Thus for any path $\ell$ from $S$ to $v$ in $\mathbf{w}_1$, there is an edge $e \in \ell$ such that $e \notin \mathbf{w}_2$. We take first such $e = (u_1, u_2)$, thus the edges on $\ell$ before $e$ are live in $\mathbf{w}_2$ and the starting node $u_1$ of $e$ is activated under $\mathbf{w}_2$ without edge $e$. Therefore there is an edge $e = (u_1, u_2)$ on the path from $S$ to $v$ such that

1. $u_1$ is activated by $\mathbf{w}_2$ on the graph without edge $e$;

2. $\mathbf{w}_1(e) = 1, \mathbf{w}_2(e) = 0$.

Such an edge $e$ is relevant with $S$ and $v$. Thus

$$\mathbb{E}_{\mathbf{w}_1,\mathbf{w}_2 \sim A}[f(S, \mathbf{w}_1, v) - f(S, \mathbf{w}_2, v)]$$
$$\leq \sum_{e \text{ is relevant with } S,v} \mathbb{E}_{\mathbf{w}_2}[\mathbb{1}\{e \text{ is observed under } \mathbf{w}_2\} \cdot (U(e) - \bar{w}(e))].$$

$\square$

With the help of this theorem, we can get the same result of TPM conditions in the work [49, 47].