[Reviews · NeurIPS 2020]

Review 1

Summary and Contributions: This paper studies the problem of online influence maximization with the LT model, rarely used in the literature compared to the IC model. An important assumption of the paper is the type of feedback: it is assumed that each step of the diffusion is observed by the learning agent. Two algorithms are proposed: one based on the well known LinUCB, and the other is a variant of the ETC algorithm. Bounds on the regret are proven for these two algorithms, as well as an experimental study (in appendix).

Strengths: This paper has the merit of tackling the setting of the LT model, much more difficult to analyze than the IC model, and arguably better to use in some context. Proven regret bounds make sense in a setting with such limited feedback. It seems that the significance/relevance of the work behind this paper is not negligible (because the technical challenges are there)

Weaknesses: My main concern for this paper is that the idea of using LinUCB in the LT model is not new (despite everything, the idea of an algorithm is not enough to prove its theoretical efficiency). I also have a small problem with the use of "node level" feedback terminology. For me, node-level feedback is simple to describe: the agent observes all the nodes influenced in the round t, period. Here we have a little more: each step of the diffusion is observed, which gives slightly more information. I'm not against this kind of setting, but I find it hard to see the added value between the whole observation of the diffusion and the simple "node-level" feedback. In particular, it seems that the setting where only the influenced nodes are observed can, in principle, benefit from a linUCB type algorithm: for each node, we can observe the Bernoulli variable of parameter the sum of the weights of the influenced incoming nodes. I think that the main issue with this kind of feedback (the "true" node-level one) is that all three terms in Theorem 1 are not covered (which is probably why the way to use your richer feedback in the algorithm is randomly chosen with probability 1/2). I may be mistaken, and I'd like the authors to give me more explanation on what I mentioned above. A dedicated paragraph in the paper would, I think, also be very helpful.

Correctness: From what I've checked, the analysis seems solid.

Clarity: This paper is otherwise pretty clear and easy to follow.

Relation to Prior Work: The various previous works seem to me quite well covered, and this paper stands out well in the contributions.

Reproducibility: Yes

Additional Feedback: My main suggestion is to provide more explanation as to the relevance and necessity of the feedback used. —— After rebuttal : I acknowledge I have read and taken into account the rebuttal


Review 2

Summary and Contributions: This paper studies online influence maximization (OIM) under the linear threshold (LT) model. OIM has hitherto been studied in the independent cascade model, and this is the first work that studies it in the LT model. The paper proposes two algorithms, a UCB-style and an explore-then-commit (ETC) style algorithm, and analyzes their regret. == after rebuttal == Thank you for your response.

Strengths: The paper is relevant to the community and explores an interesting direction for influence maximization. The theoretical claims are sound, and the contribution is novel.

Weaknesses: The paper lacks experiments. While I appreciate the theoretical contribution of the paper, it would be good to compare the empirical performance of the two proposed algorithms, and its comparison to an algorithm that chooses seeds randomly. I also found the paper a bit hard to read. It took me multiple readings to understand that E_{t,1} and E_{t,2} corresponds to the times just before the activation of the node. I feel the paper would also benefit from using terminology to denote the two notions of time (t and tau in line 156). When the authors say node level feedback, one may think that the learner observes the nodes that are activated at each time. However the feedback here is more granular.

Correctness: Yes.

Clarity: No. Please look at the weaknesses pointed out.

Relation to Prior Work: Yes.

Reproducibility: No

Additional Feedback:


Review 3

Summary and Contributions: This paper considered the online influence maximization problem based on a combinatorial MAB approach. Instead of a commonly used independent cascade model, the authors choose to use a linear threshold model, where the threshold $\theta_v$ for any node $v$ is drawn independently and uniformly in $[0,1]$. Based on this model, they propose two algorithms, an UCB based algorithm LT-LinUCB and an ETC(Explore-Then-Commit)-based algorithm OIM-ETC, and their corresponding regret analysis. They also use some experiments to demonstrate their theoretical results.

Strengths: This is an novel topic in bandit problems. The proofs of their theorems are correct and their experiments can demonstrate their theoretical results.

Weaknesses: For this paper, my major concern is on the model. In my opinion, either we follow a FREQUENCY setting to suppose that both the edge weight $e_{u,v}$ and the threshold $\theta_v$ are fixed but unknown constant in $[0,1]$, and the goal of our algorithm is to learn them out to achieve good performance; or we follow a BAYESIAN setting to suppose that both the $e_{u,v}$ and $\theta_v$ follow some known prior distribution, and we are going to minimizing the Bayesian regret of this model by some online learning policies. In this paper, the edge weight $e_{u,v}$ are fixed and unknown, while the threshold $\theta_v$ follows a uniform distribution in $[0,1]$ as its prior distribution. This is really strange. Unfortunately, I found that the algorithms in this paper highly rely on this half-frequency-half-Bayesian setting. Because of this, I think the authors need to explain and discuss more about the motivation of using a half-frequency-half-Bayesian setting, some related works about this setting and how this scenario may appear in reality.

Correctness: I check most of the proofs in the supplementary file, and I think they are correct.

Clarity: The paper is well written.

Relation to Prior Work: It is clearly discussed.

Reproducibility: Yes

Additional Feedback: ============After reading other reviews and the rebuttal================ I understand the applicability of this model by reading the rebuttal and other reviews. Therefore my score changes to "6".

[Author Response · NeurIPS 2020]

We thank the reviewers for the valuable comments and discussions. Please find our clarifications below. We
use [Narasimhan et al.'15] for Narasimhan H et al., Learnability of influence in networks, NeurIPS'2015.

- Reviewer 3: About the setting of online linear threshold model

Recall that in setting of the (offline) linear threshold (LT) model, the weight $w(e)$ associated with each edge
$e \in E$ is fixed and known and the threshold for each node is uniformly drawn from $[0,1]$. The weights are
model parameters while the thresholds are not model parameters. This setting was originally proposed in the
seminal work by Kempe et al. [19], and most follow-up studies adopt this particular setup (e.g. [9,12,16]).
The particular setting of using fixed and nonnegative weights on the edges (with the sum of weights of
the incoming edges of each node at most 1) plus the uniform sampled threshold from $[0,1]$ enables the LT
model to have an equivalent live-edge graph formulation, and unifies the LT and IC under the more general
triggering model [19], which is in turn important for deriving a number of properties (such as submodularity)
and algorithmic solutions (such as the reverse influence sampling approach [38]) for the LT model. Therefore,
our online influence maximization (OIM) is directly on the classical LT model, turning model parameters
$w(e)$ to be unknown (but fixed) and to be learned in an iterative manner. This is in parallel to the OIM for
IC model [11,43,45], which also learns unknown edge probability parameters.

It is interesting that the reviewer brought up the frequentist versus Bayesian view on OIM-LT. Per our above
discussion, we first want to clarify that the threshold on each node is not a model parameter of the classical
LT model and our work is a frequentist approach for the online setting. Alternative Bayesian approach, such
as Thompson sampling algorithm, is one of the future directions we plan to explore next. It is also possible to
analyse Bayesian regret under Bayesian setting where the weights follow some prior distributions. The offline
problem of including thresholds as model parameters, where both weights and thresholds are fixed and known,
is the fixed threshold model [9,19] and has very different property and behavior, e.g. it is not submodular
and is NP-hard to approximate to any nontrivial factor [9,19]. Also its diffusion process is deterministic for
each seed set, while under IC and LT models the diffusions are random. So how to design its online setting
and the corresponding Bayesian setting would be interesting future directions.

- Reviewer 1&2: About node-level feedback

First, we will clarify our naming by saying "full node-level feedback" as knowing the set of nodes activated at
each time step, and "partial node-level feedback" as knowing only the set of nodes activated by the end of
the diffusion process. This naming is consistent with [Narasimhan et al.'15], which also shows that these two
types of feedback give different PAC-learnability results — the full feedback allows polynomial-time learning
algorithm while the partial feedback may require an exponential-time learning algorithm.

Our paper studies the full node-level feedback. As reviewer 1 pointed out, it is possible to apply LinUCB-type
algorithm to partial node-level feedback, but the difficulty is to analyze its regret. The key to bound the
regret is to prove a similar GOM property (our Theorem 1) using the information that can be observed.
Such a Lipschitz-type property is essential since we always use estimated weights to select seeds in the online
setting and need to bound the difference caused by the estimation error. Thus, Theorem 1 is one of our main
technical contributions, and extending it to the partial node-level feedback is unclear at the moment and is
part of future research work. The probability $1/2$ (line 12 of Algorithm 1) comes from the key GOM property
(Theorem 1) and also addresses the correlation between $E_{t,1}$ and $E_{t,2}$.

- Reviewer 1: The idea of using LinUCB and the importance of LT model

(a) Yes, the idea of using LinUCB is natural, since the diffusion process involves linear structure. As we stated
above, the main difficulty is to analyze the regret for such a specific setting and feedback. (b) Although IC
may be studied in the literature more than LT, LT is still a fundamental diffusion model and various aspects
of LT has been studied (e.g. [9,12,16,19,20,38] and many other studies). For the online setting, existing
work does focus on IC with edge-level feedback [11,43,45], and this is because the independence on edge-level
propagation and edge-level feedback make the setting easier to analyze.

- Reviewer 2: Experiments and the definitions of $E_{t,1}, E_{t,2}$

(a) We have shown experiments to compare LT-LinUCB and OIM-ETC in Appendix F. Since the algorithm
of randomly selecting seeds does not learn good seed set, we did not include it. (b) For each diffusion, $E_{t,2}(v)$
denotes the set of incoming edges of active in-neighbors for node $v$ at the time when $v$ is activated; and
$E_{t,1}(v)$ denotes the set of incoming edges of active in-neighbors for node $v$ just one time step before $v$ is
activated, which are the edges just failing to influence $v$; if $v$ is not activated in the end, $E_{t,1}(v)$ is the set of
incoming edges of all its active in-neighbors, which is the largest edge set failing to influence $v$. We will add
more descriptions for better understanding of these two terms.

[Meta-Review · NeurIPS 2020]

After the rebuttal and the following discussion all reviewers agreed that the papers should be accepted and consider this as a solid submission o online influence maximization under a relevant but not well studied (due to additional challenges wrt IC) linear threshold model.